# Meteorological drought under historical and future climate scenarios in North Gojjam sub-basin, Abay River basin of Ethiopia

Tatek Belay[1]*, Tadele Melese[2], Baye Terefe[3], Simachew Bantigegn Wassie[4], Teshager Zerihun Nigussie[5]

1 Department of Geography and Environmental Studies, College of Social Science and Humanities, Debre Tabor University, Debre Tabor, Ethiopia, 2 Department of Natural Resource Management, College of Agriculture and Environmental Science, Bahir Dar University, Bahir Dar, Ethiopia, 3 Department of Geography and Environmental Studies, Injibara University, Injibara, Ethiopia, 4 Department of Geography and Environmental Studies, Faculty of Social Science, Bahir Dar University, Bahir Dar, Ethiopia, 5 Department of Statistics, College of Natural Science, Debre Tabor University, Debre Tabor, Ethiopia

* tatekbelay@gmail.com

## Abstract

The impacts of climate change are expected to vary considerably across regional and local scales, underscoring the urgent need for localized assessments. This study investigates the spatio-temporal characteristics of meteorological drought across three distinct periods- baseline (1985–2018), near future (2020–2050), and mid-future (2051–2081) under two climate scenarios: SSP2–4.5 and SSP5–8.5, within the North Gojjam sub-basin of the Abay Basin, Ethiopia. It further examines long-term trends in precipitation and both minimum and maximum temperatures across the sub-basin. Future climate projections were derived using six global climate models (GCMs) from the Coupled Model Intercomparison Project Phase 6 (CMIP6) under the Shared Socioeconomic Pathways (SSPs). Data from seven stations were extracted, bias-corrected, and aggregated using R software and relevant analytical packages. Key statistical metrics confirm a strong alignment between corrected GCM outputs and observed historical data. Meteorological drought was assessed using the Standardized Precipitation Evapotranspiration Index (SPEI) over a three-month scale, with run theory applied for drought characterization. Precipitation and SPEI trends were analyzed using the Mann-Kendall test and Sen's slope estimator, with statistical significance set at p < 0.05. Projections suggest an increase in both minimum and maximum mean annual temperatures during the near and mid-future periods, with minimum temperatures rising more sharply. Under SSP5–8.5, precipitation is expected to decrease, except during the mid-future period. The SPEI indicates an intensification and increased frequency of severe drought events. The northeastern and southeastern parts of the North Gojjam sub-basin are particularly vulnerable, posing significant risks for agriculture and water resource management. This study

which permits unrestricted use, distribution, and reproduction in any medium, provided the original author and source are credited.

**Data availability statement:** All relevant data are within the paper and its Supporting Information files.

**Funding:** The author(s) received no specific funding for this work.

**Competing interests:** The authors declare no conflict of interest.

provides critical localized insights into future climate scenarios, highlighting the importance of temporal drought forecasting and the need for targeted adaptation and mitigation strategies in the region.

## Introduction

The Earth's climate system continues to undergo unprecedented changes [1]. The observed rise in atmospheric greenhouse gas (GHG) concentrations has significantly altered climate feedback mechanisms, posing serious threats to both human societies and natural ecosystems [2]. As a result, the frequency and intensity of extreme weather events, particularly droughts, have increased markedly [3].

Drought is a common and recurring climatic phenomenon that affects all climate regimes and many regions around the world [4,5]. However, it varies in nature across space and time, reflecting unique climatic, meteorological, hydrological, and socio-economic characteristics [6,7]. Additionally, it is one of the most complicated and catastrophic climate-related phenomena and has been a significant concern for humankind for centuries [8]. They have the potential to devastate large areas and have severe consequences for agriculture [9], water resources [10], ecology [11], environment [12], and economy and society [13,14].

Among the various types of droughts, meteorological drought, which is mainly caused by a lack of precipitation, has received considerable attention, as it often serves as the precursor to other forms such as agricultural and hydrological droughts [15]. The cumulative effects of agro-meteorological and hydrological drought lead to socio-economic drought. This type of drought disrupts the entire ecosystem and causes negative impacts on human and animal lives, including potential loss of life [16].

Currently, approximately half of the Earth's land surface is susceptible to droughts [17]. Numerous studies have reported that drought events are becoming more frequent and severe globally due to the impacts of climate change [18,19]. Long-term temperature increases, coupled with unpredictable precipitation patterns, are intensifying environmental stressors. Consequently, the frequency, duration, severity, and spatial extent of drought-affected areas are projected to rise in the future, resulting in widespread and significant damage [20]. Both observed and anticipated drought intensification pose serious threats to nearly all forms of livelihood [21]. Although often difficult to quantify and frequently underestimated, droughts already account for substantial economic losses—estimated at $6.4 billion annually in the United States and €9.0 billion in the European Union. The effects are even more pronounced in less developed countries, where droughts contribute to significant loss of life and vital resources [17]. These impacts are expected to escalate further as climate change continues to evolve in complexity [22].

As outlined in the 6th Assessment Report (AR6) released by the Intergovernmental Panel on Climate Change (IPCC), developing nations are highly vulnerable to the subsequent impacts of droughts [23]. Africa, in particular, experiences more severe

droughts [24] due to limitations in technology, institutions, and finances [23]. This problem is likely to become more widespread with the increasing temperatures and declining precipitation levels in most parts of Africa. According to Masih et al. [25], there were around 291 drought events in Africa from 1900 to 2013, affecting over 362,225,799 people. Estimations indicate that over 500,000 people died in the continent during the 1980s due to recurring droughts, leading to extensive impoverishment and economic stagnation in many countries [26].

East Africa has often experienced prolonged and severe droughts due to deficiencies in electricity provision, leading to significant impacts on agricultural activities. As a result, East Africa has frequently encountered severe and extended periods of dryness, attributed to deficits in electricity supply and substantial repercussions on agricultural practices [27]. The nations within East Africa heavily depend on agriculture that relies on rainfall, an aspect greatly impacted by the shifting climate [28].

Ethiopia is one of the East African nations vulnerable to droughts. The extent and occurrence of droughts are intensifying, although some areas are more susceptible. Ethiopia has often experienced severe drought and famine [29]. Meteorological and agricultural droughts occurred on average every two years between 1950 and 2017, with 34 droughts occurring between these years [30]. Out of these, 13 were severe, affecting the entire country [29]. As a result of these changes, people in the country become more susceptible, and their livelihoods are disrupted, resulting in long-term or short-term food insecurity [31]. Because of this, natural resources are being destroyed when they could be used for development instead. Despite the frequent recurrence of droughts, there is a lack of well-defined strategies to mitigate their impact.

Generally, drought affects nearly every region of Ethiopia, including areas that were previously unaffected [30]. However, northeastern and southeastern parts, rift valley regions, and the Upper Blue Nile basin including some parts of the Amhara regions mainly South Wollo, North Wollo, and South Gondar are affected more frequently than the others [32–34].

The main focus of this study, the North Gojjam sub-basin, has been identified as one of the most drought-vulnerable areas in the country. Specifically, the South Gonder zone, part of the North Gojjam sub-basin, has been classified as one of the country's most drought-vulnerable regions. In the area, persistent droughts have posed a significant risk to the sustenance of rural communities and the stability of food supplies. In most parts of the area, crop failure and development efforts are threatened by localized drought disasters. Therefore, there is a need for accurate monitoring and assessment of meteorological drought disasters to improve their management and early warning systems.

Drought monitoring and forecasting studies remain unexplored and require a comprehensive investigation to mitigate drought-related socio-economic problems. Although drought is the most significant source of uncertainty for farming households, it has received little scientific attention, with no attempts to quantify the meteorological and agricultural drought's spatio-temporal characteristics in the study area. Moreover, recent studies indicate that climate change could lead to extreme temperatures, unusual rainfall events, and more strong, extended droughts [34–36]. The anticipated changes in precipitation patterns, both spatial and temporal, may trigger new drought characteristics in affected areas. Inaccurate information about the spatiotemporal characteristics that cause droughts can lead to poor decision-making, increasing the costs and damages associated with droughts [37]. As a result, a better understanding of the spatiotemporal trends of droughts, particularly agricultural droughts, is essential for early detection, effective monitoring, mitigation efforts, and minimizing socio-economic losses.

Understanding how droughts change over space and time is crucial, especially given ongoing climate change and increased human activities. Therefore, it is essential to acquire reliable predictions of future drought for developing effective mitigation and adaptation strategies. Climate projections serve as the foundation for future drought assessments. Numerous studies based on forecasts from Global Climate Models (GCMs) indicate that the frequency, duration, intensity, and severity of droughts have increased globally [38,39]. Recent advancements in the Coupled Model Intercomparison Project Phase 6 (CMIP6) have produced updated climate model outputs, offering more precise predictions of extreme weather events as many regions experience increasing dry trends. Analysis of climate model experiments offers valuable

insights into how climate change affects drought, particularly through multi-model efforts. This study utilizes climate projections from the Coupled Model Intercomparison Project Phase 6 (CMIP6) to quantify global changes in meteorological drought under two future warming scenarios: the Shared Socioeconomic Pathways SSP2–4.5 and SSP5–8.5.

Numerous studies have been conducted in Ethiopia on meteorological and agricultural droughts, only a few of them have focused on predicting future meteorological droughts. For example, Bayissa et al. [37] used the SPI to analyze drought characteristics in the Upper Blue Nile Basin. Gidey et al. [40] also utilized SPI to predict meteorological drought hazards in Northern Raya Valley under medium emission scenarios. Most recently, Yisehak et al. [41] used the Reconnaissance Drought Index (RDI) to investigate spatio-temporal patterns of meteorological drought in the semi-arid highlands of northern Ethiopia, considering both high and medium emission scenarios. However, to overcome the limitations of SPI and RDI, it is essential to assess past and future meteorological droughts in Ethiopia based on SPEI under both SSP2–4.5 and SSP5–8.5 emission scenarios. Zhao et al. [42] noted that SPEI, incorporating temperature data, is an ideal index for examining the impact of climate change on model outputs under various future scenarios.

The objective of this study is to analyze the trends, frequency, duration, and intensity of meteorological droughts in the North Gojjam sub-basin, taking into account climate variability during the baseline period and projected changes under medium- and high-emission scenarios. In addition, the study seeks to examine the spatiotemporal variations and trends in both rainfall and temperature across the sub-basin. The insights gained from this research can inform strategies to enhance water resource management, strengthen regional climate forecasting, and improve the resilience and livelihoods of communities in the North Gojjam sub-basin and other similarly affected regions.

## Materials and methods

### Description of the study area

The study was carried out in the Ethiopian highlands, North Gojjam sub-basin of the Abay River basin, located between 10°00′ to 11°45′ N and 37°15′ to 38°30′ E. Administratively, it is situated in the Amhara regional state, bordering the North Gojjam, East Gojjam, and South Gondar Zones. The sub-basin is one of the major headstreams of the Abay River, accounting for about 8% of the Abay River basin's area and contributing around 14% of its total annual flow [43]. Additionally, the Upper Blue Nile River receives a large amount of sedimentation from the sub-basin [44]. Its elevation ranges from 1,009–4,094 meters above sea level, covering an area of approximately 14390 km$^2$ (Fig 1).

The average annual rainfall of the study area varies from 930 mm in the southeast region to 1575 mm in the west region. Between 1985 and 2018, the average annual minimum and maximum temperatures were 10.1 °C and 24.7 °C, respectively. According to the CSA [45] of Ethiopia, the area has a population density of 210.6 persons per km$^2$ with an average farm size of less than 1 ha per household.

The study area's climatic seasons are defined as follows: the dry season from October to February, locally called *Bega*; a minor rainfall period from March to May, known locally as *Belg*; and the long rainy period from June to September (summer), locally called *Kiremt*, with the heaviest rainfall occurring between July and August. Rainfall in the area is primarily influenced by the movement of the inter-tropical convergence zone, which is driven by moisture from the Indian Ocean, the equatorial East Pacific, the Gulf of Guinea, the Mediterranean region, and the Arabian Peninsula [46]. Moreover, human activities have significantly impacted the hydrological cycle in the study area.

For the local population, the main sources of income are raising cattle and cultivating crops. The main cultivated crops include wheat, barley, teff, maize, millet, and sorghum. Eucalyptus plantations are expanding as a source of alternative revenue, competing for land formerly used for annual crops [47]. The topographic and climate gradient in the area allows for the cultivation of crops from both tropical and temperate origins. The main soil types found are Nitisols, Vertisols, Luvisols, Leptosols, Fluvisols, and Cambisols [48].

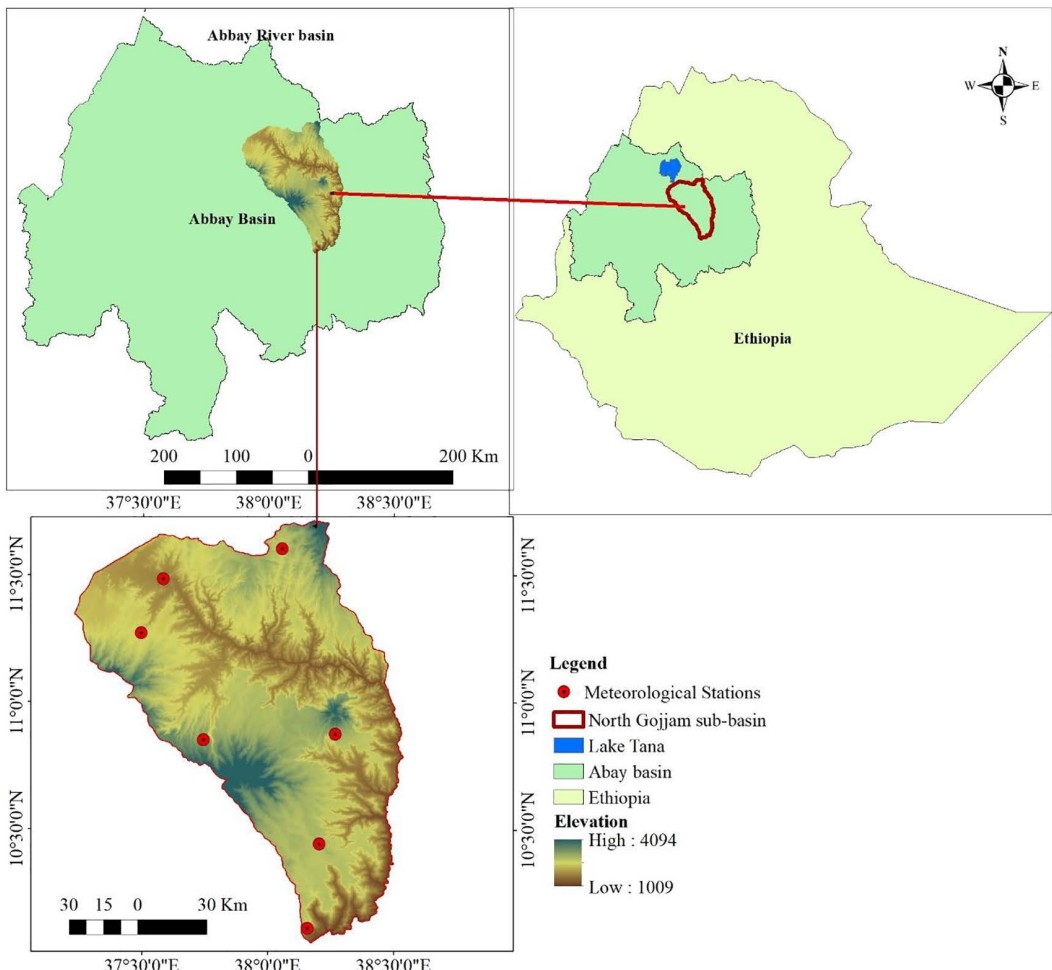

**Fig 1. Map of the study area developed by the authors.** Source: Original shapefile data freely downloaded from https://public.opendatasoft.com/.

## Data and methods

**Datasets. Observed data:** The Ethiopian Meteorological Institute provided the daily records of gridded precipitation and the dataset for daily minimum and maximum temperatures, which have a spatial resolution of 4 km x 4 km. The dataset covers the period from 1985 to 2018. To extract the sub-basin time series data, seven meteorological stations situated in and around the sub-basin were used. The data is used as the baseline for bias correction of CMIP6 models and to analyze trends and assess historical meteorological droughts.

**GCMs baseline and future data:** Accurate future climate projections are routinely derived using a range of outputs and scenarios from Global Climate Models (GCMs) [49]. The variables included in downscaling climate data from GCMs can vary based on the specific research/application. However, this study focused on temperature variables (minimum and maximum) and precipitation. In this study, we utilized Six- Global Climate Models (GCMs): ACCESS-CM2, BCC-CSM2-MR, EC-Earth3-CC, MIROC6, MPI-ESM1–2-LR, and NorESM2-LM. This study also utilized data available from the website portal at https://aims2.llnl.gov/search/cmip6/. The data generated from these models were used to project future drought indices in the study area from 2020 to 2081. The choice of these GCM models is based on previous research conducted in different regions of Ethiopia [50–53].

The data were extracted from grid cells covering the area, including both historical (1990–2014) and future (2020–2050) and (2051–2081) periods to ensure a fair comparison. Data extraction was performed in R using the ncdf4, raster, terra, stars, and sf packages. The list of six GCMs used in this study is presented in Table 1. These GCMs have been extensively used in previous studies [50–54] and provide projections for all three selected scenarios throughout the entire study period.

According to the 6th IPCC (2021) report, there are five emission scenarios (Shared Socioeconomic Pathways (SSPs). These scenarios combine socio-economic trends with greenhouse gas emissions: SSP1–1.9, SSP1–2.6, SSP2–4.5, SSP3–3.7, and SSP5–8.5. This study focuses solely on SSP2–4.5 and SSP5–8.5, representing mid-range and high-level climate scenarios, to provide a comprehensive range for climate analysis by 2100. These scenarios serve as a foundation for climate modeling and impact assessments, helping to understand the potential range of climate change outcomes based on varying emission trajectories.

## Methods

### Statistical bias correction and validation

Data from Global Circulation Models (GCMs) or Regional Circulation Models (RCMs) may exhibit biases [55]. Due to the accompanying systematic and random model flaws, bias correction of GCM output is essential [56–58]. Hence, bias correction methods are employed to correct the output from GCMs with observed data. To reduce potential biases between observed and simulated climate variables, it is necessary to correct both historical and GCM scenario runs using a transformation algorithm before their usage. This ensures the accuracy and reliability of the data.

Numerous bias correction methods have been widely used for precipitation, including power transformation, quantile mapping (QM), de-trend quantile mapping (DQM), and quantile delta mapping (QDM). Meanwhile, linear scaling, delta change correction, and variance scaling are used as temperature bias correction methods. In this study, QM-based bias correction was used for precipitation, while linear scaling bias correction methods were used for temperature. These methods were chosen because they are relatively simple to implement and have shown effectiveness in correcting biases in climate data. They are commonly employed to adjust climate model data and improve accuracy for impact assessments. We used gridded rainfall and temperature data provided by NMA to correct data from six Global Climate Models (GCMs) from the Coupled Model Inter-comparison Project phase 6 (CMIP6) project. Bias correction was performed in RStudio using the Qm package prior to conducting the drought analysis. Precipitation data were adjusted with multiplier terms, while temperature data were corrected with additive terms.

**Table 1. List of six selected CMIP6 climate models for climate projection in North Gojjam sub-basin.**

| No | GCMs | Institute |
|---|---|---|
| 1 | ACCESS-CM2 | Commonwealth Scientific and Industrial Research Organisation, Australia |
| 2 | BCC-CSM2-MR | Beijing Climate Center, China |
| 3 | EC-Earth3-CC | European research institutions and universities |
| 4 | MIROC6 | University of Tokyo, Japan Meteorological Agency, and National Institute for Environmental Studies, Japan |
| 5 | MPI-ESM1–2-LR | Max Planck Institute for Meteorology (MPI-M), Germany |
| 6 | NorESM2-LM | Norwegian Climate Centre (NorESM), Norway |

## Evaluation of the accuracy of bias correction

Evaluating the performance of a bias corrected data is essential for accurate drought prediction. Previous studies have demonstrated that several authors assessed prediction models using statistical standards. This study evaluated the performance of bias-adjusted models using four statistical indicators: the Nash-Sutcliffe efficiency coefficient (NSE), percent bias (PBIAS), the correlation coefficient (r), and root mean square error (RMSE).

## Determination of SPEI and trend analysis

**Calculation of Standardized Precipitation-Evapotranspiration Index (SPEI).** To monitor meteorological drought, meteorological variables or parameters may be used. The associated indices are typically based on precipitation and temperature data series. These indices help quantify the intensity, duration, and frequency of drought conditions in a region. They provide valuable insights for assessing the severity of water shortages, as well as for drought prediction and early warning systems. There are various meteorological drought indices, such as the well-known Standardized Precipitation Index (SPI) and the more recently developed Standardized Precipitation Evapotranspiration Index (SPEI). A highly referenced indicator in academic literature, the SPI is one of the most commonly employed metrics for monitoring meteorological droughts [59–61]. However, by relying solely on precipitation as the input, the SPI does not consider the temperature component, which is crucial to evaluate total water balance and water usage of a region.

In this study, we used the SPEI [62] to assess drought characteristics in the study area. Numerous studies have confirmed the suitability of this index for evaluating droughts, as the SPEI accounts for the impact of potential evapotranspiration (PET) on drought intensity [63–65].

Drought in a region has been tracked and predicted using either a single indicator or multiple indices. Furthermore, hybrid indices have recently been proposed to combine the advantages of multiple data sources while maintaining a single index for decision-makers [66]. These hybrid indices, which often combine temperature and precipitation, provide a more comprehensive assessment of drought conditions. This approach addresses the limitations of relying on a single metric, which might not capture all aspects of drought severity. Recognizing that temperature and precipitation records or projections are more effective in gauging the severity of meteorological droughts, we chose to employ the SPEI index with a short time frame (3 months) for this study.

The SPEI is a drought metric calculated by standardizing the cumulative climatic water balance over a specific period. It considers both precipitation (P) and potential evapotranspiration (PET). Several methods can be used to calculate PET, but the Hargreaves and Samani [67] formula is used for PET calculation in this study. The Eq. (1) for estimating Hargreaves PET:

$$PET = 0.0023 * R_a * (T_{max} - T_{min})^{0.5} * \left( \frac{T_{max} + T_{min}}{2} + 17.8 \right)$$

(1)

Where the variable PET represents the potential evapotranspiration in millimeters. $R_a$ is extraterrestrial radiation in MJ m$^{-2}$ per day [68]. $T_{max}$ and $T_{min}$ represent maximum and minimum temperatures (°C), respectively.

The SPEI, developed by Vicente-Serrano et al. [62], is a standardized value of the difference $D_i$ between precipitation ($P_i$) and potential evapotranspiration (PET) for month i. It is calculated using the following Eq. (2).

$$D_i = P_i - PET_i$$

(2)

This method provides a straightforward measure of water surplus or deficit for the month under analysis. The $D_i$ values can be aggregated over different timescales, however, the $D_i$ are first standardized using a three-parameter log–logistic distribution (Eq. 3):

$$f(x) = \frac{\beta}{\alpha} \left( \frac{x-y}{\alpha} \right)^{\beta-1} [1 + (\frac{x-y}{\alpha})^{\beta}]^2 \qquad (3)$$

where $\alpha$, $\beta$, and $\gamma$ are calculated from probability weighted moments and used to determine the log-logistic distribution, which is applied to the $D_i$ dataset. The SPEI is then calculated as the probability exceeding a given value of $D_i$ ($W = -2 \times \ln(p)$) (Eq. 4):

$$SPEI = W - \frac{C_0 + C_1 W + C_2 W^2}{1 + d_1 w + d_2 w^2 + d_3 w^3} \qquad (4)$$

The constarnts used are: $C_0 = 2.515517$, $C_1 = 0.802853$, $C_2 = 0.010328$, $d_1 = 1.432788$, $d_2 = 0.189269$, and $d_3 = 0.001308$. For a detailed description of the SPEI calculation method, refer to the references [62,69,70].

Positive SPEI values indicate above-average moisture conditions, while negative values represent drier than normal conditions. A drought event is defined when the SPEI value is less than or equal to −1 in a given period.

SPEI values were calculated using a three-month time scale. The corresponding drought categories based on SPEI values are presented in Table 2.

**Frequency, duration, and intensity of droughts.** The study used the equations indicated in Table 3 to compute the area affected by droughts, as well as the duration, frequency, and intensity of historical drought events and the projected ones.

**Trend analysis.** Various methods, including statistical and rank-based tests, can be employed to detect rainfall and temperature patterns [73]. Examples of statistical techniques include Sen's slope estimator, least squares linear

**Table 2. The classification schemes of drought intensity [71].**

| SPEI Value | Class |
|---|---|
| >2.0 | Extremely wet |
| 1.5 to 2 | Severely wet |
| 1.0 to 1.5 | Moderately wet |
| 0.5 to 1.0 | Mild wet |
| −0.5 to 0.5 | Normal |
| −1.0 to −1.5 | Moderate drought |
| −1.5 to −2.0 | Severely drought |
| <−2.0 | Extremely drought |

**Table 3. Computation of duration, frequency, and intensity of droughts.**

| Drought Parameter | Equation | Symbol and units |
|---|---|---|
| Drought duration | $D = \dfrac{\sum_{i=1}^{n} d_i}{n}$ | D = drought duration (months) <br> di = duration of i$^{th}$ drought event <br> n = total number of drought events |
| Drought frequency | $F = \dfrac{n_m}{N_m} * 100$ | F = drought frequency (%) <br> $n_m$ = number of months <br> $N_m$ = total number of months |
| Drought intensity | $I = \left\| \dfrac{1}{n} \sum_{i=1}^{n} SPEI_i \right\|$ | I = drought intensity (-), <br> n = number of drought occurrences in months with SPEI < −1 <br> $SPEI_i$ = SPEI value below the threshold (−1) |

Source: Haile et al. [72].

regression, and slope-based tests. Rank-based methods, such as the Spearman rank correlation test and the Mann-Kendall (MK) test, are also commonly used for trend analysis.

The Mann-Kendall test, developed by Mann and Kendall between 1945 and 1975 [74,75]. is one of the most widely used methods for analyzing trends in hydrological and meteorological time series [76]. A key advantage of this test is that it does not assume any specific statistical distribution, making it a robust and non-parametric method [77]. The test is based on two hypotheses: the null hypothesis ($H_0$), which states that the data are randomly distributed with no trend, and the alternative hypothesis ($H_1$), which indicates the presence of a monotonic trend in the data series. The Mann-Kendall test can be evaluated both computationally and graphically [78].

The Mann-Kendall method is a non-parametric statistical test commonly used for detecting trends in time series data, especially in hydrology and climatology. The MK test [79,80] is a widely recognized nonparametric test used for identifying trends in meteorological variables.

This study assessed whether trends in precipitation, meteorological drought, and minimum and maximum temperature across the study area exhibited a monotonic upward or downward change using the Mann-Kendall (MK) trend test and Sen's slope estimator. Moreover, Sen [81]. demonstrated that simple nonparametric approaches are effective for evaluating the magnitude of trends in time series data. The test statistic S is calculated using the following equation (Eq. 5 and Eq. 6):

$$S = \sum_{i=1}^{n} \sum_{j=1}^{i-1} sign(x_i - x_j)$$

(5)

$$sign(x_i - x_j) = \begin{cases} +1, & if(x_i - x_j) > 0 \\ 0, & if(x_i - x_j) = 0 \\ -1, & if(x_i - x_j) < 0 \end{cases}$$

(6)

Here, $x_i$ and $x_j$ represent the values of consecutive data points, ranked from $i = 1, 2, \ldots, n-1$, where n covers the entire period. The original variance was calculated using Eq. (7) as follows:

$$var(s) = \frac{n(n-1)(2n+5)}{18}$$

(7)

A trend may emerge when autocorrelation is present over a given period. Therefore, in situations where autocorrelation exists, it is recommended to use the standardized Mann-Kendall approach.

The new variance of S, calculated using this improved technique, is shown in Eq (8):

$$var(s) = \left[ \frac{n(n-1)(2n+5)}{18} \right] x \frac{n}{n_e^*}$$

(8)

Where $\frac{n}{n_e^*}$ represent the autocorrelation correction factor. It is calculated as follows in Eq. (9):

$$\frac{n}{n_e^*} = 1 + \frac{2}{n(n-1)(n-2)} \sum_{i=1}^{n-1} (n-i) \, x \, (n-i-1) \, x (n-i-2) \, x \, p(i)$$

(9)

P(i) symbolizes the autocorrelation function of the ranks of the time series.

To account for the influence of serial correlation on trend detection, we applied a lag-1 autocorrelation correction following the standard approach in climate studies. Lag-1 autocorrelation correction adjusts for the correlation between consecutive observations in a time series. This adjustment is critical for trend analyses, such as the Mann–Kendall test, because uncorrected autocorrelation can inflate the apparent significance of a trend, leading to false positives.

The MK Statistic (Z) obtained by Eq. (10):

$$Z = \begin{cases} if (S < 0)\ then\ \frac{S+1}{\sqrt{var(s)}} \\ if (x_i - x_j) > \ then\ \frac{S-1}{\sqrt{var(s)}} \\ if (x_i - x_j) = 0\ then\ 0 \end{cases}$$

(10)

A p-value of less than or equal to 0.05 indicates a statistically significant trend, with the direction determined by the sign of the test statistic, while a p-value greater than 0.05 suggests the absence of a statistically significant trend [82].

The Sen's Slope Estimator is a non-parametric technique used to determine the strength of trends in SPEI. Eq. (11) calculates the Sen's slope as follows:

$$s_i = \frac{x_j - x_k}{j - k}\ i = 1, ..., k$$

(11)

Where $x_j$ and $x_k$ are the values of time series at times j and k (j>k), respectively. The sum of squares about the median (SSE) is determined by calculating the median of N values of $S_i$. If the SSE is positive, it suggests an upward trend in the time series, while a negative SSE indicates a downward trend. The overall methodology employed in this study is summarized in Fig 2.

## Results

### The spatiotemporal trend of precipitation and temperature

**Trend analysis of rainfall, minimum temperature, and maximum temperature from 1985 to 2018.** The MK trend test and Sen's slope were used to investigate the study area's monthly maximum and minimum temperature data [84]. The study area exhibited significant variation in mean annual rainfall across the study area. Digo Tsion recorded the highest mean annual rainfall of 1380.6 mm, while Bichena recorded the lowest mean value of 957.70 mm. There are notable differences in rainfall patterns within the study sub-basin due to the heterogeneous distribution of rainfall. The study area's meteorological stations showed an increasing trend in mean annual rainfall, as shown in Table 4, even though this trend was not statistically significant. However, the stations at Digo Tsion, Kuy, and Mekane Eyesus showed statistically significant upward trends in annual rainfall. This aligns with earlier studies conducted in the Tana sub-basin [85,86], which also found similar evidence of an increasing trend in mean annual rainfall. Conversely, the meteorological station at Adet exhibited a downward trend during the study period. The magnitude of the annual rainfall trend varies geographically, ranging from −0.63 to 11.13 mm/year.

From 1985 to 2018, the mean annual minimum temperature in the study area ranged from 8.2°C to 12.6°C, while the mean annual maximum temperature ranged from 20.19 to 28.65 °C. These results indicate significant variations in the mean annual minimum and maximum temperatures across the study area. The findings suggest that Tis Abay station had a considerably greater mean annual maximum temperature at 28.64 °C, while Digo Tsion had the lowest mean annual minimum temperature at 8.2 °C.

There has been an upward trend in the mean annual minimum and maximum temperatures observed at most meteorological stations within the study sub-basin (S1, S2 and S3 Figs). This increment was statistically significant for all stations (p<0.05) except Kuy in the minimum temperature and Adet station in the maximum temperature. According to Sen's

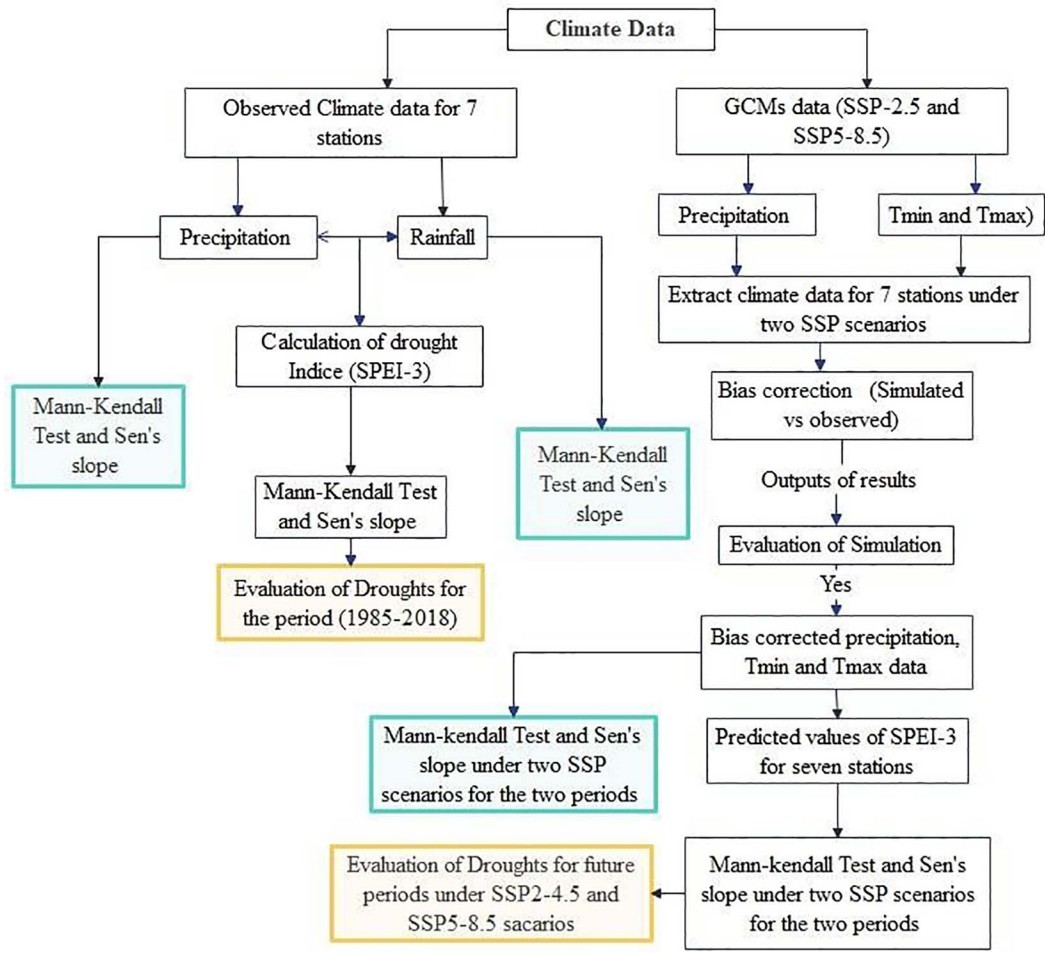

**Fig 2. Flowchart of the methodology applied in this study (Adopted from [83].**

**Table 4. Mann-Kendall (M-K) test and Sen's Slope values for Rainfall, minimum temperature (Tmin), and maximum Temperature (Tmax) for the 1985-2018.**

| Stations | Rainfall | | | Minimum Temperature | | | Maximum Temperature | | |
|---|---|---|---|---|---|---|---|---|---|
| | *Zmk* | *Sen slope* | *P<0.05* | *Zmk* | *Sen slope* | *P<0.05* | *Zmk* | *Sen slope* | *P<0.05* |
| Adet | −0.24 | −0.63 | 0.813 | 4.80 | 0.06 | 0.000 | 1.58 | 0.01 | 0.114 |
| Bichena | 1.90 | 4.35 | 0.058 | 3.06 | 0.05 | 0.002 | 1.95 | 0.02 | 0.051 |
| Digo Tsion | 2.24 | 8.64 | 0.025 | 4.20 | 0.04 | 0.000 | 3.05 | 0.03 | 0.002 |
| Kuy | 1.13 | 6.96 | 0.015 | 1.35 | 0.02 | 0.176 | 3.27 | 0.03 | 0.001 |
| Mekane eysus | 3.05 | 11.13 | 0.002 | 2.98 | 0.03 | 0.002 | 3.12 | 0.02 | 0.002 |
| Merto lemariam | 0.79 | 2.43 | 0.458 | 3.22 | 0.03 | 0.001 | 4.44 | 0.03 | 0.000 |
| Tis Abay | 0.98 | 3.33 | 0.327 | 4.94 | 0.06 | 0.000 | 3.31 | 0.03 | 0.001 |

slope analysis, the rate of increment in minimum temperatures ranged from 0.02°C to 0.06°C over the past three decades over the study area (Table 4). The annual rate of increase in maximum temperature ranged from 0.01°C to 0.03°C in the study sub-basin. These findings imply that the annual rate of increase in minimum temperature is greater than that of the

maximum temperature in the study area over the last three decades. According to earlier studies [87–89], Ethiopia's mean annual temperature has been increasing. The data in Table 4 supports the conclusion that there is a noticeable trend of rising temperatures across the stations. Several studies [90,91] have referred to the last three decades as "the warmest in the history of the planet Earth."

**Evaluation of bias correction methods.** The statistical metrics indicate that the modeled precipitation, minimum temperature, and maximum temperature were successfully adjusted to align with observed data, despite raw simulations previously being underestimated. Statistical calculations were employed to evaluate each model's performance.

Under the SSP2–4.5 and SSP4–8.5 climate scenarios for the mid and future periods, the minimum and maximum temperatures and precipitation levels in seven towns demonstrated satisfactory model performance. Notably, the predicted values for these towns closely matched the observed data, as evidenced by correlation coefficients (R) approaching 1. The Percent Bias (PBIAS) and Root Mean Square Error (RMSE) metrics further confirmed strong model performance, with PBIAS values near zero and relatively low RMSE values. These results indicate a high level of agreement between the observed and downscaled temperature and precipitation data. Although the PBIAS, RMSE, and R values varied slightly among the towns, attributable to differences in temperature extremes and precipitation variability, the overall findings consistently reflected a reliable and accurate representation of the observed climatic conditions.

The resulting statistical values suggest that the model's performance in downscaling GCMs' minimum and maximum temperatures and precipitation was consistent across the seven stations, despite their geographical differences. Then, using the bias-corrected data, the trends of the predicted minimum temperature, maximum temperature, and precipitation characteristics for two future periods (2020–2050 and 2051–2081) under the SSP2–4.5 and SSP5–8.5 scenarios were examined. The data was also used to investigate the duration, severity, and frequency of droughts in the North Gojjam sub-basin.

**Minimum and maximum temperature and rainfall changes under future climate scenarios for the study area.** Climate projections for the study region indicate that mean annual precipitation, as well as mean minimum and maximum temperatures, will vary over time and across different scenarios. In the near future, under the SSP2–4.5 scenario, the projected mean annual minimum and maximum temperatures are 12.13°C and 25.83°C, respectively, while the average annual precipitation is estimated at 991.07 mm. Under the SSP5–8.5 scenario for the same period, precipitation is expected to be higher, at 1064.47 mm, accompanied by slightly increased mean annual minimum and maximum temperatures of 12.21°C and 26.02°C, respectively. These findings suggest that SSP5–8.5 may lead to marginally warmer and wetter conditions compared to SSP2–4.5.

In the mid-term period (2051–2081), both models predict increased precipitation and warmer temperatures. Under the SSP2–4.5 scenario, mean annual precipitation is projected to rise to 1048.45 mm, while mean annual maximum and minimum temperatures are expected to reach 26.79°C and 13.08°C, respectively. Meanwhile, the SSP5–8.5 scenario forecasts higher temperatures, with mean annual maximum and minimum temperatures of 27.18°C and 13.94°C, respectively. Precipitation in this scenario is also predicted to be higher, at 1136.5 mm. These results highlight how the SSP5–8.5 scenario reflects the intensifying effects of rising greenhouse gas emissions.

**Analysis of rainfall and temperature trends for the period 2020–2050.** The analysis of rainfall data depicted spatial variations across the study sub-basin. The spatial variations of the study sub-basin were illustrated by analyzing rainfall data. The findings of the Mann-Kendall trend test reveal that, under SSP2–4.5, there will be a statistically significant positive trend in mean annual precipitation at various meteorological stations, except at the Kuy station. Conversely, a statistically significant increase in the trends of minimum and maximum temperatures is expected at most of the study sub-basin stations.

The analysis indicates that the greatest increases in minimum and maximum temperatures are expected between 2020 and 2050 under the SSP5–8.5 and SSP2–4.5 scenarios. The most substantial upward trends are expected in both minimum and maximum temperatures within the timeframe of 2020–2050 under the SSP5–8.5 and SSP2–4.5 scenarios.

The lowest predicted minimum temperature increase is expected at Tis Abay, while the highest increase is expected at Mekane eysus (0.061 °C/year). This is followed by Adet and Digo Tsion, which are expected to increase at a rate of 0.035 and 0.032 °C/year, respectively. The rate of maximum temperature under near future scenario (2020–2050) under SSP2–4.5 is also expected to increase at a significant level at all stations, and the rate of change ranges from 0.017 °C per year (Digo Tsion) to 0.022 °C per year (Mekane eysus) (Table 5). The study findings indicate spatial variations in rainfall, as well as in minimum and maximum temperatures, across the study area under near-future climate scenarios (S4–S9 Figs).

Rainfall trends in the study sub-basin during the mid- and near-future periods under the SSP5–8.5 scenario exhibit a positive trajectory. However, these changes are statistically insignificant, as indicated by the Mann-Kendall analysis (Table 6). In contrast, both minimum and maximum temperatures are projected to increase significantly across the study area in the near future. Bichena and Merto Lemariam are expected to experience the smallest annual increase in minimum temperature, at a rate of 0.045°C per year, while Adet is projected to have the highest annual increase, at 0.053°C per year. For maximum temperature, the SSP5–8.5 scenario predicts increases from 2020 to 2050, ranging from 0.031°C per year in Bichena and Kuy to 0.037°C per year in Mekane Eyesus (Table 6). These findings suggest that minimum temperatures are rising at a faster rate than maximum temperatures across the study area.

### Analysis of rainfall and temperature trends for the period 2051–2081

According to the study's findings, the minimum temperature in Merto Lemariam is projected to increase by 0.063 °C per year under the SSP2–4.5 scenario. Notably, the minimum temperature is rising at a faster rate than the maximum temperature across the study area. Furthermore, Mekane Eyesus is expected to experience the highest annual increase in maximum temperature under this scenario, at 0.016 °C per year (Table 7). Under the SSP5–8.5 scenario, Kuy and Tis Abay are projected to exhibit higher rates of increase in both minimum and maximum temperatures during the 2060s, with annual increases of 0.074 °C and 0.059 °C, respectively (Table 8; S10–S15 Figs).

**Table 5. Mann-Kendall test and Sen's Slope values for Rainfall, Tmin, and Tmax from 2020–2050 under SSP2-4.5.**

| Stations | Rainfall | | | Minimum Temperature | | | Maximum Temperature | | |
|---|---|---|---|---|---|---|---|---|---|
| | $Z_{mk}$ | Sen slope | P<0.05 | $Z_{mk}$ | Sen slope | P<0.05 | $Z_{mk}$ | Sen slope | P<0.05 |
| Adet | 2.583 | 4.855 | 0.009 | 5.370 | 0.035 | 0.000 | 4.718 | 0.019 | 0.000 |
| Bichena | 1.257 | 2.375 | 0.208 | 4.792 | 0.029 | 0.000 | 5.222 | 0.020 | 0.000 |
| Digo Tsion | 2.787 | 5.809 | 0.005 | 5.234 | 0.032 | 0.000 | 4.459 | 0.017 | 0.000 |
| Kuy | −0.509 | −0.647 | 0.610 | 4.826 | 0.031 | 0.000 | 5.200 | 0.021 | 0.000 |
| Mekane eysus | 2.141 | 4.210 | 0.032 | 6.492 | 0.061 | 0.000 | 3.843 | 0.022 | 0.000 |
| Merto lemariam | 1.053 | 3.485 | 0.291 | 4.996 | 0.026 | 0.000 | 4.946 | 0.020 | 0.000 |
| Tis Abay | 2.277 | 3.723 | 0.022 | 4.130 | 0.016 | 0.000 | 4.848 | 0.018 | 0.000 |

**Table 6. Mann-Kendall test and Sen's Slope values for Rainfall, Tmin, and Tmax from 2020–2050 under SSP5-8.5.**

| Stations | Rainfall | | | Minimum Temperature | | | Maximum Temperature | | |
|---|---|---|---|---|---|---|---|---|---|
| | *Zmk* | *Sen slope* | *P<0.05* | *Zmk* | *Sen slope* | *P<0.05* | *Zmk* | *Sen slope* | *P<0.05* |
| Adet | 2.311 | 6.005 | 0.020 | 5.302 | 0.053 | 0.000 | 4.623 | 0.035 | 0.000 |
| Bichena | 1.121 | 2.372 | 0.231 | 5.472 | 0.045 | 0.000 | 4.351 | 0.031 | 0.000 |
| Digo Tsion | 1.223 | 2.779 | 0.221 | 5.608 | 0.049 | 0.000 | 4.436 | 0.032 | 0.000 |
| Kuy | 1.427 | 2.925 | 0.153 | 5.472 | 0.048 | 0.000 | 4.317 | 0.031 | 0.000 |
| Mekane eysus | 1.631 | 3.957 | 0.102 | 5.370 | 0.051 | 0.000 | 4.691 | 0.037 | 0.000 |
| Merto lemariam | 0.305 | 0.480 | 0.759 | 5.608 | 0.045 | 0.000 | 4.521 | 0.032 | 0.000 |
| Tis Abay | 2.515 | 3.431 | 0.011 | 5.574 | 0.046 | 0.000 | 4.691 | 0.035 | 0.000 |

**Table 7. Mann-Kendall test and Sen's Slope values for Rainfall, Tmin, and Tmax from 2051 to 2081 under SSP2-4.5.**

| Stations | Rainfall | | | Minimum Temperature | | | Maximum Temperature | | |
|---|---|---|---|---|---|---|---|---|---|
| | Zmk | Sen's slope | P | Zmk | Sen's slope | P | Zmk | Sen's slope | P |
| Adet | 0.339 | 1.09 | 0.73 | 4.776 | 0.022 | 0.000 | 4.037 | 0.013 | 0.000 |
| Bichena | −0.101 | −0.32 | 0.91 | 5.506 | 0.024 | 0.000 | 3.713 | 0.012 | 0.000 |
| Digo Tsion | 0.645 | 1.62 | 0.51 | 4.792 | 0.022 | 0.000 | 4.070 | 0.013 | 0.000 |
| Kuy | 0.441 | 0.58 | 0.65 | 5.472 | 0.026 | 0.000 | 3.616 | 0.012 | 0.000 |
| Mekane eysus | 0.713 | 1.39 | 0.47 | 4.894 | 0.023 | 0.000 | 3.162 | 0.016 | 0.000 |
| Merto lemariam | 0.475 | 2.10 | 0.63 | 6.118 | 0.063 | 0.000 | 3.454 | 0.013 | 0.000 |
| Tis Abay | 0.645 | 1.05 | 0.51 | 6.322 | 0.062 | 0.000 | 3.778 | 0.015 | 0.000 |

**Table 8. Mann-Kendall test and Sen's Slope values for Rainfall, Tmin, and Tmax from 2051 to 2081 under SSP5-8.5.**

| Stations | Rainfall | | | Minimum Temperature | | | Maximum Temperature | | |
|---|---|---|---|---|---|---|---|---|---|
| | $Z_{mk}$ | Sen's slope | P | $Z_{mk}$ | Sen's slope | P | $Z_{mk}$ | Sen's slope | P |
| Adet | 1.087 | 3.491 | 0.276 | 6.050 | 0.070 | 0.000 | 5.982 | 0.049 | 0.000 |
| Bichena | 1.733 | 4.510 | 0.082 | 6.016 | 0.071 | 0.000 | 5.710 | 0.047 | 0.000 |
| Digo Tsion | 1.733 | 4.901 | 0.082 | 6.118 | 0.069 | 0.000 | 5.880 | 0.045 | 0.000 |
| Kuy | 1.665 | 4.483 | 0.095 | 6.016 | 0.074 | 0.000 | 5.778 | 0.046 | 0.000 |
| Mekane eysus | 0.339 | 1.004 | 0.733 | 6.492 | 0.061 | 0.000 | 5.982 | 0.057 | 0.000 |
| Merto lemariam | 1.155 | 2.957 | 0.247 | 6.118 | 0.063 | 0.000 | 5.914 | 0.047 | 0.000 |
| Tis Abay | 0.339 | 1.037 | 0.733 | 6.322 | 0.062 | 0.000 | 6.204 | 0.059 | 0.000 |

**Drought trends based on 3-month SPEI under different climate scenarios in the North Gojjam sub-basin.** The 3-month SPEI trends for seven stations were analyzed over three time periods: the baseline period (1985–2018), and two future periods (2020–2050 and 2051–2081), under the SSP2–4.5 and SSP5–8.5 climate scenarios. The Mann-Kendall (MK) test and Sen's slope estimator were employed to detect and quantify temporal trends at each station. The results revealed distinct temporal and spatial variations in drought patterns across the study area, influenced by differing greenhouse gas emission scenarios.

During the baseline period, significant increasing moisture trends were observed at Digo Tsion, Kuy, and Mekaneeyesus, as indicated by the Mann-Kendall (MK) test ($p < 0.05$) and corresponding positive Sen's slope values (0.0025, 0.0017, and 0.003, respectively). These findings suggest a shift toward wetter conditions at these stations. In contrast, other stations such as Adet, Bichena, and Tis Abay did not exhibit statistically significant trends, underscoring localized variability in climate patterns during this period.

During the mid-century period (2020–2050), the SSP2–4.5 scenario exhibited significant increasing moisture trends across most stations. Adet, Bichena, Digo Tsion, Mekane Eyesus, and Tis Abay all recorded positive Sen's slope values ranging from 0.001 to 0.002, indicating a shift toward wetter conditions that may help mitigate drought risks and improve water availability in the region. Under the SSP5–8.5 scenario, significant moisture trends were observed at fewer stations, specifically Adet, Mekane Eyesus, and Tis Abay. Although positive trends were still recorded, the spatial extent of statistically significant changes was more limited compared to SSP2–4.5, suggesting more localized impacts potentially driven by higher emission intensities.

For the far-future period (2051–2081), no statistically significant trends were detected at any station under either the SSP2–4.5 or SSP5–8.5 scenarios ($p > 0.05$). Sen's slope values were close to zero or negative, indicating either stabilization or a slight drying trend. These results suggest that although significant moisture increases are projected for the mid-century period, such trends may not continue into the latter half of the century. The observed stabilization may reflect

a complex interplay between rising temperatures and changes in precipitation patterns driven by broader global climate dynamics.

The findings have important implications for regional water resource management and agricultural planning. Projected increases in moisture trends during the mid-century period, particularly under the SSP2–4.5 scenario, may enhance water availability and reduce drought risk in key areas such as Adet and Mekane Eyesus. However, the variability observed under SSP5–8.5 highlights the need for adaptive strategies tailored to localized conditions. The stabilization of trends in the far-future period underscores the necessity of accounting for long-term uncertainties in climate projections, as ecosystems may approach the limits of their adaptive capacity under a sustained high-emission scenario.

Table 9 illustrates the temporal distribution of 3-month SPEI trends across the seven stations in the North Gojjam sub-basin for the baseline and future periods under both SSP2–4.5 and SSP5–8.5 scenarios. The data indicate that with increasing emission intensity, particularly under SSP5–8.5, Sen's slope values generally increased during the mid-future period, reflecting more pronounced trends toward wetter conditions. However, for the far-future period, stabilization was evident across all stations, with no significant increases or decreases in SPEI trends.

The near-future projections under SSP2–4.5 indicated more consistent moisture trends across all stations, while SSP5–8.5 exhibited localized significant trends, primarily in Adet, Mekane Eyesus, and Tis Abay. These results underscore the complex interactions of temperature, precipitation, and emission intensities in shaping future drought patterns in the North Gojjam sub-basin.

## Analysis of drought frequency, duration, and intensity under different climate scenarios

To examine trends in the duration of SPEI-3 drought events. this study applies run theory to identify and characterize drought events based on their duration and severity. Orginally proposed by Yevjevich [92], run theory is a well established method for detecting the longest sequences of consecutive drought months under drought consitions. Run theory is one of the most effective methods for analyzing time series data. It focuses on the identification of different event types within a continuous sequence of similar events, such as droughts, consecutive rain-free days, periods of rainfall, or other alternating hydrological conditions.

Time series analyses were used to capture temporal variations in drought characteristics. The results indicate a gradual increase in both the frequency and intensity of droughts over time. This upward trend in the length and persistence of drought periods suggests a heightened likelihood of prolonged drought events in a warming climate. By analyzing sequences of consecutive drought months, we quantified the frequency and average duration of drought episodes across different timeframes. Specifically, the study assessed drought patterns under two climate scenarios- SSP2–4.5 and SSP5–8.5 across three distinct periods: historical (1985–2018), near future (2020–2050), and mid-future (2051–2081). The findings reveal substantial spatial variability among meteorological stations, highlighting the importance of region-specific adaptation strategies to effectively mitigate future drought impacts.

**Table 9. Sen's slope values for 3-month SPEI trends at seven stations under the baseline, SSP2-4.5, and SSP5-8.5 scenarios.**

| Scenarios | Adet | | Bichena | | Digo Tsion | | Kuy | | Mekane Eyesus | | Merto lemariyam | | Tis Abay | |
|---|---|---|---|---|---|---|---|---|---|---|---|---|---|---|
| | P | Sen's slope | P | Sen's slope | P | Sen's slope | P | Sen's slope | P | Sen's slope | P | Sen's slope | P | Sen's slope |
| 1985-2018 | 0.414 | 0.000 | 0.227 | 0.001 | 0.000 | 0.003 | 0.000 | 0.002 | 0.000 | 0.003 | 0.001 | 0.001 | 0.541 | 0.000 |
| 2020-2050 (SSP2–4.5) | 0.001 | 0.001 | 0.005 | 0.001 | 0.000 | 0.001 | 0.660 | 0.000 | 0.000 | 0.002 | 0.042 | 0.001 | 0.000 | 0.002 |
| 2020-2050 (SSP5–8.5) | 0.010 | 0.001 | 0.132 | 0.001 | 0.075 | 0.001 | 0.057 | 0.001 | 0.029 | 0.001 | 0.990 | 0.000 | 0.001 | 0.001 |
| 2051-2081 (SSP2–4.5) | 0.856 | 0.000 | 0.722 | 0.000 | 0.734 | 0.000 | 0.969 | 0.000 | 0.783 | 0.000 | 0.592 | 0.000 | 0.745 | 0.000 |
| 2051-2081 (SSP5–8.5) | 0.741 | 0.000 | 0.843 | 0.000 | 0.802 | −0.001 | 0.868 | 0.000 | 0.729 | 0.000 | 0.889 | 0.000 | 0.840 | 0.000 |

**Drought during the baseline period (1985–2018)**

The study reveals substantial spatial and temporal variability in meteorological drought characteristics across seven monitoring stations over the past three decades (1985–2018). These differences are evident in the frequency, duration, and intensity of drought events, all of which have significant implications for agricultural productivity and water resource management in the region. Meteorological droughts, defined by extended periods of below-average precipitation, are a primary driver of these challenges. As shown in Fig 3, all seven stations recorded periods of rainfall deficit during the study period. However, the magnitude and persistence of these deficits varied markedly across both time and location, underscoring the localized nature of drought impacts and the necessity for tailored mitigation strategies.

The study findings revealed that Tis Abay experienced the highest number of drought episodes (112), with the shortest average duration of 2.46 months. This indicates that droughts occurred frequently but lasted for shorter periods. In contrast, Mekane Eyesus experienced the lowest number of drought events (94) but had the longest average duration, at 3.12 months, suggesting fewer but more prolonged dry spells. Adet and Mertolemariam fell between these extremes, with 108 and 98 drought events, respectively, and average durations of 2.61 and 2.69 months.

At Adet, meteorological droughts exhibited considerable variability in duration, ranging from short episodes lasting 1–3 months to extended dry periods of up to 12 months. Short-duration droughts dominated, accounting for approximately 70% of all events. However, prolonged droughts, particularly those lasting more than five months, were relatively uncommon. Notably, the severe droughts that occurred between August to October 2016 and from July to September 2017 were exceptionally intense, with SPEI values of −1.493 and −1.482, respectively.

By the end of the baseline period, these extreme events pointed to an emerging trend in the northern parts of the study area, particularly in Adet, toward more severe, albeit shorter, drought episodes. While rare, such events underscore a broader shift toward increasing rainfall variability and growing unpredictability in the regional climate.

The analysis reveals that various parts of the study region experienced meteorological droughts with notable differences in frequency, severity, and duration. For instance, Bichena faced recurrent droughts, including significant multi-month events occurring in March. A particularly severe drought, lasting six months and peaking in March 2018, reached an SPEI intensity of −1.751 due to substantial precipitation deficits. These prolonged droughts exacerbated water scarcity and disrupted local agricultural systems, highlighting the region's increasing vulnerability to extended dry spells in recent years.

Similarly, Digo Tsion experienced multiple extreme drought events, the most severe occurring between August and November 1989 (SPEI=−1.760). Other extended drought periods were observed in 1986–1987, 1992–1993, and 2009–2010, many persisting for over six months. The intensity of these events, often exceeding −1.0 on the SPEI scale, severely affected agricultural productivity and water availability in the area.

Kuy also endured several severe droughts, including a major event from July to November 1989 (SPEI=−1.523). Prolonged droughts lasting more than five months occurred during 1986–1987, 1992–1993, and 2009–2010, contributing to persistent agricultural stress and ecosystem disruption.

Mekane Eyesus was notably impacted by long-lasting droughts. The longest event spanned 17 months, from January 1992 to May 1993 (SPEI=−1.137), while the most extreme occurred between July 1989 and January 1990 (SPEI=−1.584). Additional significant droughts were recorded in 1986, 1997, 2005, and 2011, with many events lasting over six months.

In Mertolemariam, drought durations varied widely, ranging from brief but intense episodes to prolonged dry spells. Extended droughts of up to 14 months were recorded during 1996–1997, 2004–2005, and 2009–2010. One of the most severe episodes occurred between April and June 1990, reaching an SPEI of −1.623.

Tis Abay also experienced frequent and extended droughts, particularly during the late 1980s, 1992–1993, and the 2000s. Notable events include a 12-month drought from July 2004 to June 2005 (SPEI=−1.067) and a seven-month

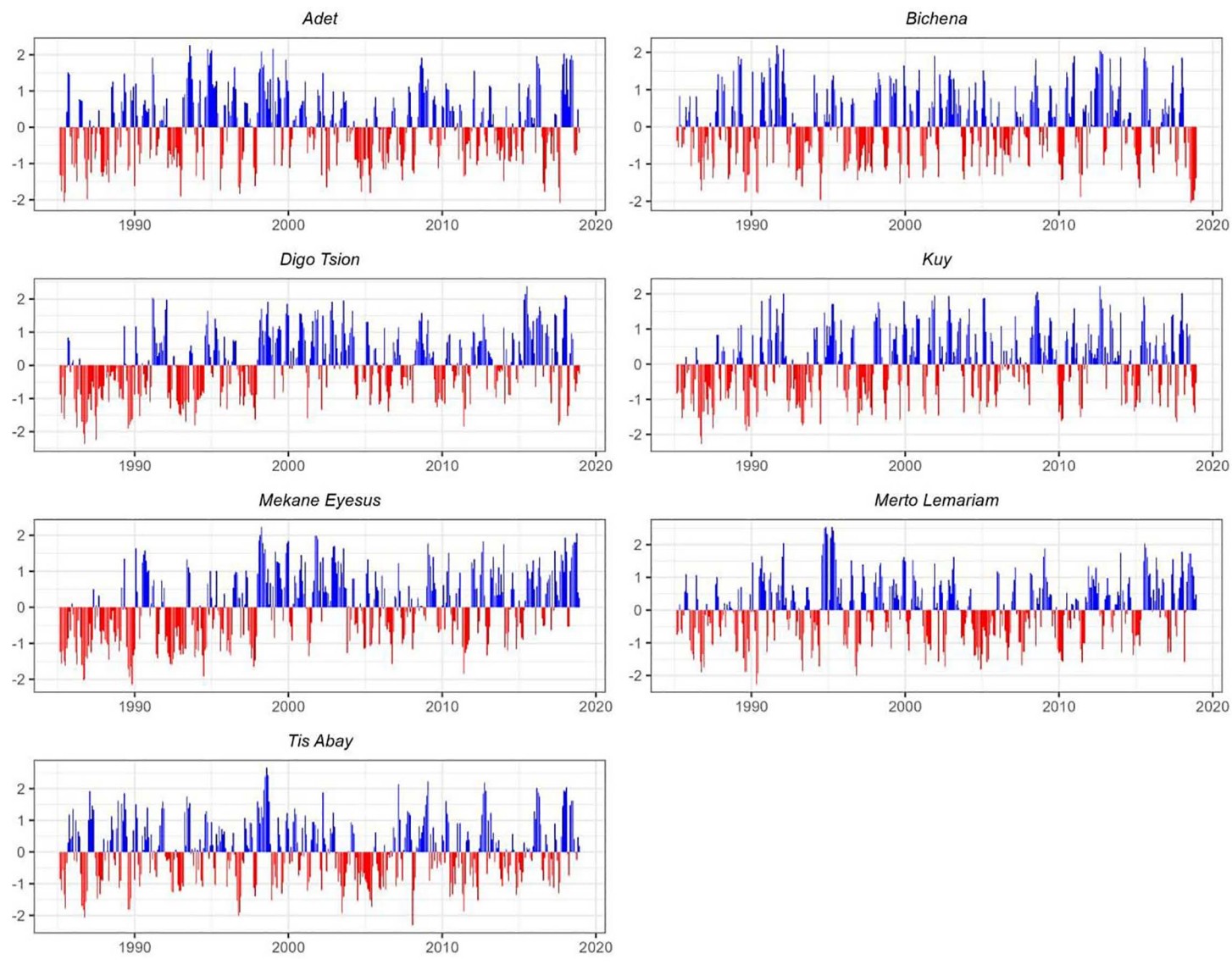

**Fig 3. SPEI values over a 3-month timescale for seven locations between 1985 and 2018.**

drought between 2003 and 2004 (SPEI = −1.111). These were compounded by severe episodes in 1986–1987 and 1992–1993, as illustrated in Fig 3.

Overall, the analysis of drought events from 1985 to 2018 indicates a combination of both short- and long-duration droughts, with significant regional variations in intensity. While short-duration droughts have become more frequent in recent years, long-duration droughts during the 1980s, 1990s, and early 2000s posed substantial challenges to agricultural output and water security. These findings underscore the urgent need for adaptive strategies, including enhanced water storage infrastructure, climate-resilient agricultural practices, and responsive early warning systems to address increasing drought variability and build long-term resilience in the region.

## Drought trends during 2020–2050 under the SSP2–4.5 scenario

An evaluation of drought conditions at seven stations in the North Gojjam subbasin from 2020 to 2050 under the SSP2–4.5 scenario predicts an increase in both drought frequency and intensity in the near future, particularly during the early to mid-2020s, 2030s, and 2040s. Locations such as Adet, Bichena, Digo Tsion, Kuy, and Mekane Eyesus are projected to experience frequent and severe drought events, underscoring the urgent need for enhanced water resource management strategies.

The findings of the study indicate that droughts in Adet are expected to last between one and nine months, with the most severe droughts occurring between 2020 and 2025. The longest drought is expected to extend from May 2022 to January 2023, with an intensity of −1.25. Additionally, peak drought events in 2025–2026 are forecasted to reach an intensity of −1.52. These results highlight the urgent need for enhanced water management strategies in Adet and the northern parts of the study sub-basin.

Severe droughts are also anticipated in Bichena during 2020, 2025–2026, and the early 2030s. The most severe drought, with a SPEI value of −1.67, is expected between December 2025 and April 2026. This event reinforces the importance of developing long-term drought mitigation strategies, as droughts lasting 7–9 months are likely to recur during the 2030s.

In Digo Tsion, a severe drought with an intensity of −1.72 is projected for the period between December 2025 and March 2026. Situations are expected to deteriorate further throughout the 2040s, culminating in an extreme drought event in 2048 with an intensity of −2.26. Kuy is forecasted to experience droughts ranging from one to fifteen months, with the longest lasting from 2030 to 2032. The most intense drought, reaching −1.99 on the SPEI scale, is also predicted for 2048. In Mekane Eyesus, multiple severe droughts are anticipated throughout the 2030s and 2040s. Notably, a 15-month drought is expected from September 2031 to November 2032, alongside an event in 2048 with an intensity of −1.99 (Fig 4).

Overall, the analysis predicts that from 2020 to 2050, droughts will become more frequent and severe across the region, with durations ranging from a few months to over a year. This trend underscores the critical need for effective water resource management, climate-resilient agricultural practices, and the implementation of robust early warning systems to mitigate the escalating impacts of drought.

## Drought during the period 2020–2050 under the SSP5–8.5 scenario

This study investigated meteorological drought occurrences at seven locations- Adet, Bichena, Digo Tsion, Kuy, Mekane Eyesus, Merto Lemariyam, and Tis Abay, between 2020 and 2050 under the SSP5–8.5 climate scenario. It highlights the droughts' duration, intensity, and seasonal patterns across the research sub-basin, aiming to provide valuable insights for future agricultural planning and water resource management. Table 10, illustrates the spatiotemporal variations in drought duration and frequency across the seven stations within the study area.

According to the analysis, the SSP5–8.5 scenario predicts that Adet will experience the highest frequency of droughts (106 episodes) during the near-future period, with relatively short average durations of 2.28 months. Conversely, Merto Lemariyam will experience shorter drought episodes (88 episodes) but a longer average duration of 2.84 months. This inverse relationship between drought frequency and duration underscores the need for professional adaptation strategies, as it demonstrates the varied impacts of drought across different locations.

At Adet, droughts are projected to occur in all seasons, with short- and medium-term events being particularly frequent. Short-term droughts lasting one to two months, such as those in June 2021, October 2022, and July 2026, will occur regularly. Medium-term droughts spanning three to five months, like those from July to November 2020 and August to November 2025, will also be common. While long-term droughts lasting six months or more will be rare, they will have significant impacts, as evidenced by events from April to September 2034 and March to October 2041. Drought intensity is expected

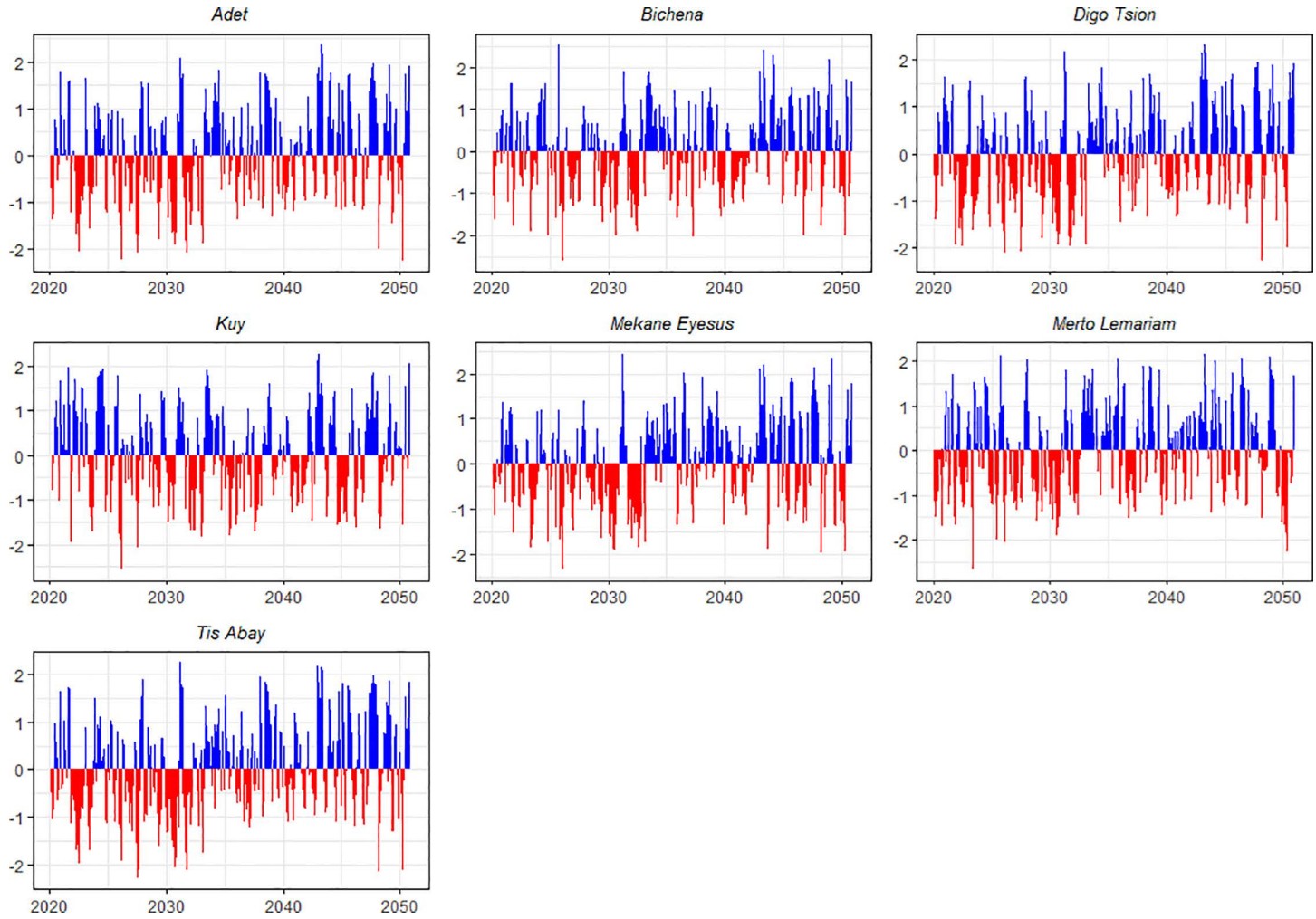

**Fig 4. SPEI trends over a 3-month timescale for seven locations between 2020 and 2050 under SSP2-4.5.**

**Table 10. Duration and Frequency of drought events based on the 3-month SPEI across three time periods (1985–2018, 2020–2050, and 2051–2081) at seven stations under two SSP scenarios.**

| Stations | Drought frequency | | | | | Duration | | | | |
|---|---|---|---|---|---|---|---|---|---|---|
| | 1985-2018 | 2020-2050 | | 2051-2081 | | 1985-2018 | 2020-2050 | | 2051-2081 | |
| | | SSP2–4.5 | SSP5–8.5 | SSP2–4.5 | SSP5–8.5 | | SSP2–4.5 | SSP5–8.5 | SSP2–4.5 | SSP5–8.5 |
| Adet | 108 | 100 | 106 | 94 | 92 | 2.61 | 2.40 | 2.28 | 2.48 | 2.67 |
| Bichena | 96 | 84 | 96 | 96 | 76 | 2.83 | 2.85 | 2.56 | 2.36 | 3.368 |
| Digo Tsion | 98 | 100 | 94 | 100 | 72 | 2.77 | 2.36 | 2.4 | 2.58 | 3.47 |
| Kuy | 98 | 92 | 96 | 88 | 76 | 2.87 | 2.56 | 2.35 | 2.77 | 3.26 |
| Mekane eyesus | 94 | 78 | 98 | 104 | 88 | 3.12 | 3.12 | 2.51 | 2.44 | 2.7 |
| Mertolemariyam | 98 | 110 | 88 | 112 | 82 | 2.69 | 2.25 | 2.84 | 2.35 | 3.26 |
| Tis Abay | 112 | 92 | 94 | 106 | 100 | 2.46 | 2.58 | 2.4 | 2.35 | 2.48 |

to range from mild to severe, with severe droughts recorded in 2020, 2036, and 2041. The summer and fall months will see the highest concentration of severe droughts, including the most extreme event, an eight-month-long drought in 2041. These findings highlight the increasing unpredictability of droughts in Adet, which will have critical implications for agriculture and water resource management.

Short-term droughts, particularly severe ones lasting one month, are expected to be common in Bichena, such as the event in July 2040. Several medium-term droughts, lasting several months, are also projected, including those from September to November 2020 and August to December 2042. Long-term droughts will be rare but significant, with one notable example being the six-month drought from September 2027 to February 2028. Drought severity will range from mild to severe, with severe droughts recorded between 2021 and 2040. Drought events are expected to cluster during the summer and autumn seasons, with 2034 experiencing the most severe event. These findings indicate a growing pattern of severe and prolonged droughts, posing serious threats to the agricultural sector in the southern regions of the research area, including Bichena.

According to climate projection results under the SSP5–8.5 scenario, Digo Tsion is expected to experience droughts in all seasons, with short-term events occurring more frequently in 2020, 2022, and 2040. Medium-term droughts lasting three to five months are projected to be common, particularly in 2020 and 2026 (Fig 5). Long-term droughts, such as the one in 2037–2038, will be significant despite their rarity. Drought intensity is expected to vary, with the most severe event projected for 2042. Seasonal clustering is anticipated, especially during the summer and fall, with some droughts spanning multiple seasons, such as the one in 2038–2039. These projections underscore the need for adaptive strategies to effectively manage water and agricultural resources during prolonged droughts.

According to the study, similar trends of increasing drought frequency and intensity are projected for the Kuy, Mekane Eyesus, Merto Lemariyam, and Tis Abay regions. Short-term droughts will be frequent, while medium-term droughts lasting several months are expected to be particularly common during the mid to late 2020s. Long-term droughts will be rare but will have significant impacts, especially in Tis Abay and Merto Lemariyam. The severity of droughts is expected to vary, with some months experiencing exceptionally harsh conditions. While some droughts are projected to occur during spring and autumn, they are anticipated to be more frequent in the summer and autumn seasons.

Drought frequency, intensity, and duration are projected to increase substantially across all stations when examining drought occurrences from 2020 to 2050 under the SSP5–8.5 climatic scenario. The results will highlight diverse drought impacts, illustrated by the inverse relationship between frequency and duration at locations such as Adet and Merto Lemariyam. These changes are expected to significantly affect water resource management, ecological resilience, and agriculture. With the increasing prevalence of both short-term and long-term droughts, proactive, region-specific adaptation strategies will be crucial to mitigating the impacts of these changes.

## Drought for the period 2051–2081 under the SSP2–4.5 scenario

The SSP2–4.5 scenario, which analyzed drought occurrences in several areas between 2051 and 2081, reveals substantial increases in drought frequency, intensity, and duration. The results for this period under SSP2–4.5 show a clear shift in precipitation patterns, consistent with trends observed in the near-term timeframe, with significant implications for agriculture, water resources, and environmental management. Droughts are expected to occur frequently, with varying characteristics across different locations.

Merto Lemariyam remains the most drought-prone area, with 112 anticipated occurrences, while Kuy is projected to experience the fewest droughts (88). Despite their high frequency, Merto Lemariyam and Tis Abay stations are expected to have the shortest average drought duration (2.35 months), whereas Kuy is projected to experience the longest average duration (2.77 months). Moderate drought conditions are anticipated for areas like Adet and Digo Tsion, with event frequencies ranging from 94 to 100 and average durations of 2.58 months. The results of the analysis indicate that a few years, specifically 2052, 2059, 2065, and 2079, are expected to stand out for experiencing multiple droughts in a single

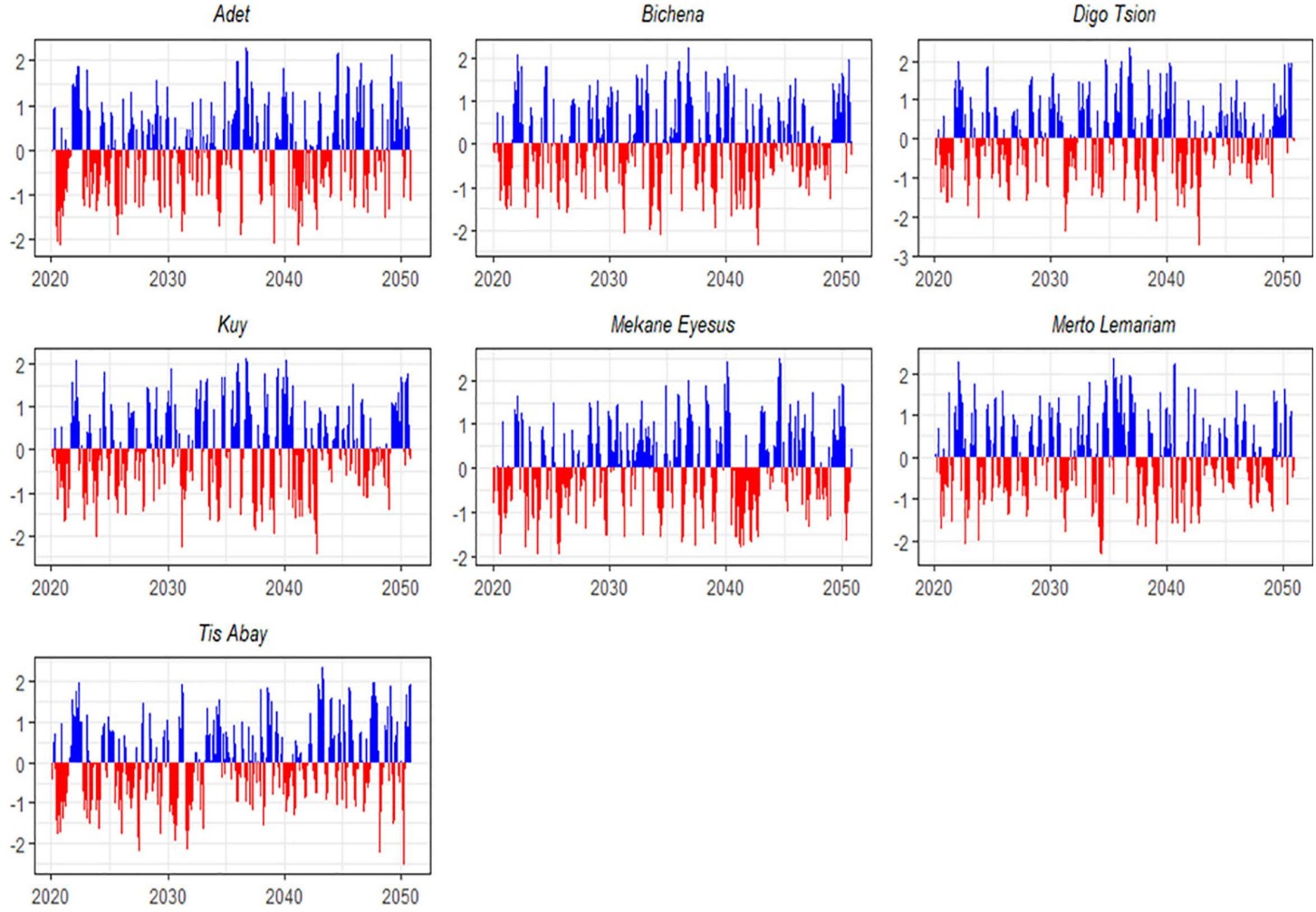

**Fig 5. SPEI trends over a 3-month timescale for seven locations between 2020 and 2050 under SSP5-8.5.**

year throughout the majority of the study area. The most likely reason for this is more unpredictable precipitation patterns, which lead to more frequent dry spells and may be detrimental to human life and local ecosystems.

The results of the study also indicated that drought spells are expected to last for one to two months. However, many persistent droughts are anticipated to endure for three to eight months, with some possibly lasting up to ten months. As indicated in Fig 6, the most serious extended drought is predicted for 2069–2070 at the Mertolemariyam station. Prolonged droughts are particularly prevalent in the early 2050s, mid-2060s, and late 2070s, underscoring the growing challenge of adapting to extended water scarcity. Such prolonged dry spells pose a major threat to agriculture, water supply systems, and ecological stability.

It is also anticipated that the intensity of drought will vary greatly, with some occurrences potentially reaching extremely high levels. In January 2053, the Kuy station is predicted to experience the worst drought on record, with an intensity of −2.09. Similarly, a very severe one-month drought with an intensity of −2.30 is predicted for Mekane Eyesus in June 2076. According to the SSP2–4.5 scenario, there will be more extreme occurrences in the 2070s and 2080s, indicating growing climatic variability. These extreme droughts will negatively impact agricultural productivity and water availability, making affected areas more vulnerable to the effects of climate change.

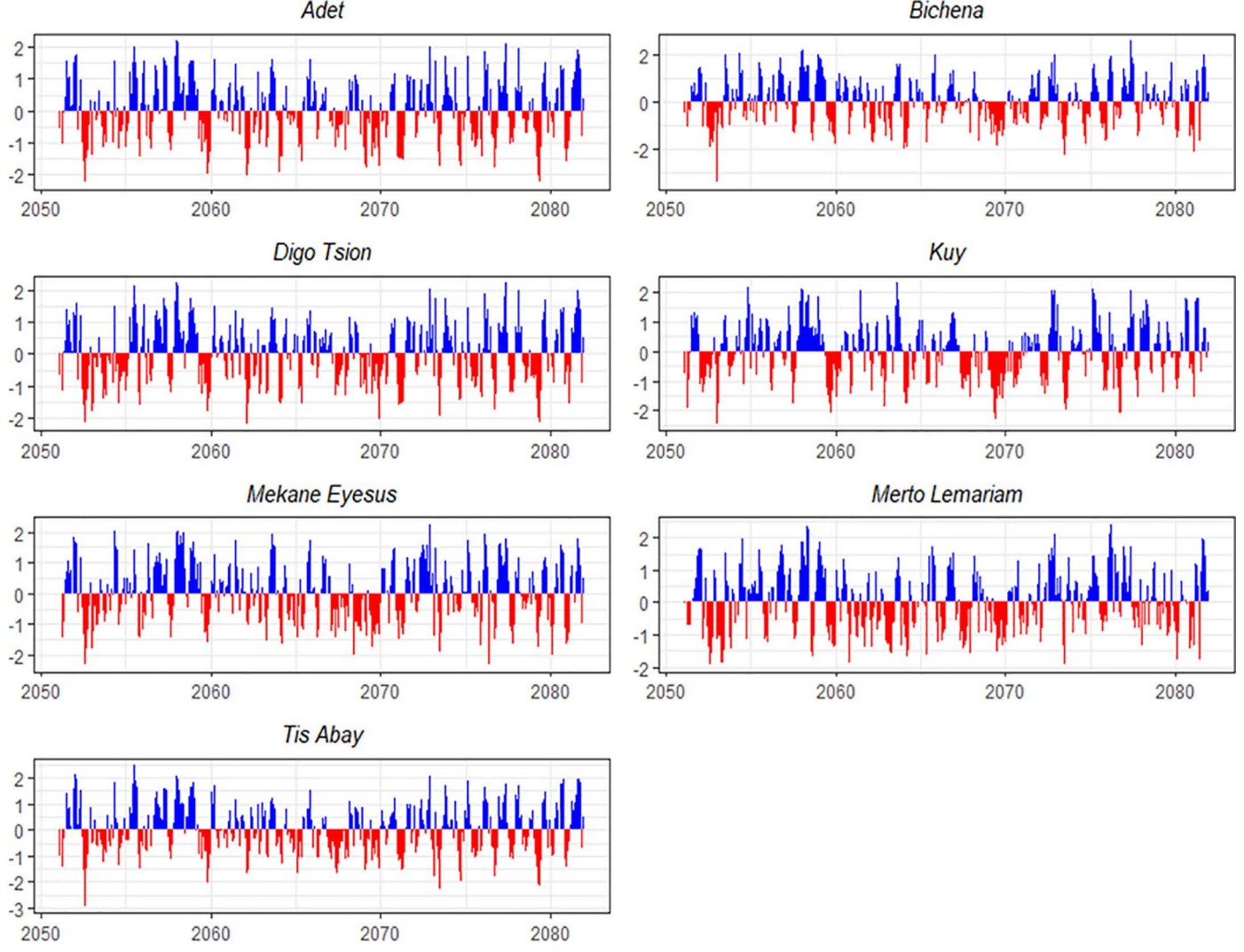

**Fig 6. SPEI trends over a 3-month timescale for seven locations between 2051 and 2081 under SSP2-4.5.**

According to the results of climate data outputs, a distinct downward trend is likely to be observed in the temporal distribution of drought episodes under SSP2–4.5 for the years 2051–2081. While droughts are predicted to occur more frequently in the 2050s, they will become considerably more common and severe in the 2060s and 2070s. Droughts are expected to last longer during these decades, with multiple occurrences in a single year. Severe, prolonged droughts will be particularly prevalent in the 2070s and 2080s under the SSP2–4.5 scenario, highlighting the increasing intensity of climate change-induced stress.

Despite the general trend of severe and frequent droughts, there are notable regional variations. Mertolemariyam and Tis Abay are predicted to experience shorter drought durations but more frequent occurrences, while Kuy is expected to have the fewest droughts but the longest average duration. Moderate drought frequencies and durations are anticipated for Adet and Digo Tsion. Early in 2053, Kuy is predicted to experience the worst drought, while in June 2076, Mekane Eyesus is expected to face the worst drought in a single month. These variations highlight the need for regional approaches to address the diverse impacts of droughts on agriculture and water supplies for the period 2051–2081.

## Drought for the period 2051–2081 under SSP5–8.5 scenario

The SSP5–8.5 scenario was used to evaluate meteorological droughts at seven locations between 2051 and 2081: Mekane Eyesus, Adet, Bichena, Digo Tsion, Kuy, Mertolemariyam, and Tis Abay. The findings indicate that drought episodes are occurring more frequently and with greater intensity. The intensity ratings of these droughts ranged from moderate to severe, with the most severe events decreasing to below −1.0. The recurrent droughts experienced at each station, often concentrated in specific years, suggest a gradual increase in dry periods over time.

According to the SPEI value analysis, Tis Abay is expected to have the shortest average drought duration (2.48 months) and the highest drought frequency (100 episodes) during the mid-term period (2051–2081). Digo Tsion and Bichena, on the other hand, are projected to experience fewer drought events (72 and 76 episodes, respectively) but with longer average durations (3.47 and 3.37 months). It is anticipated that, over time, droughts at these sites will become more frequent and severe.

The findings of the study indicate that, under the SSP5–8.5 scenario, drought frequency and intensity are projected to increase at all sites over the mid-term period. In Adet, droughts are expected to become more frequent and severe. Between 2051 and 2054, several droughts are projected, including a severe event from October to December 2055 (intensity: −1.36). A more intense drought is predicted for 2059, with the trend continuing through 2056–2060. The most severe event is anticipated between December 2076 and June 2077 (intensity: −0.98), followed by even more extreme drought in 2078 (intensity: −1.66).

Furthermore, the analysis indicates that Bichena is expected to experience severe droughts lasting 11 months, starting in July 2051 (intensity: −1.27), December 2054 (−1.10), June–December 2055 (−1.33), and March-May 2076 (−1.52). In Digo Tsion, frequent and prolonged droughts are anticipated, including a severe six-month drought in 2071 (−1.39), a nine-month event from 2068 to 2069 (−1.18), and a 13-month drought beginning in May 2051 (−1.21). Kuy is projected to face several extended droughts as well, such as a 12-month drought in 2068–2069 (−1.12), an 11-month event in 2051 (−1.24), and a severe one-month drought in March 2072 (−1.63). Similar trends of increasing drought frequency and intensity over time are expected in Mekane Eyesus and Mertolemariyam (Fig 7). In general, the predictions suggest that the stations analyzed exhibit a clear trend toward more frequent and severe droughts, highlighting the region's increasing vulnerability to water shortages and climate change.

Over the mid-future period, the analysis highlights a concerning trend of increasing drought frequency and severity at the seven stations under the SSP5–8.5 scenario. Tis Abay is expected to experience the highest frequency of droughts, while Digo Tsion and Bichena will face fewer but longer-lasting events. These findings underscore the escalating risks posed by climate change in the region, emphasizing the urgent need for effective drought management strategies and adaptation measures in vulnerable areas.

## Discussion

Historically, Ethiopia has frequently been affected by severe droughts, often accompanied by famine and exceptionally dry years [29]. These drought episodes have caused significant environmental degradation, impacted communities, and disrupted livelihoods [31]. Numerous studies have confirmed that climate change could result in extreme temperatures, irregular rainfall patterns, and more frequent, prolonged, and intense droughts [32,34,36]. This study also confirms that meteorological droughts in the North Gojjam sub-basin are projected to change significantly and intensify in the future compared to the historical period. Variations in temperature and precipitation under future climate scenarios are expected to alter the characteristics of drought events. As a result, under the SSP2–4.5 and SSP5–8.5 climate scenarios, the frequency and duration of droughts are anticipated to vary across the seven sites within the North Gojjam sub-basin. Moreover, climate change is projected to increase drought severity. However, the frequency, duration, and intensity of droughts will likely vary depending on the specific climatic conditions and historical periods being considered.

**Fig 7. SPEI trends over a 3-month timescale for seven locations between 2051 and 2081 under SSP5-8.5.**

The trend analysis of mean annual rainfall, monthly maximum temperature, and minimum temperature in the study area during the historical period (1985–2018) was conducted using the Mann-Kendall (MK) trend test. The results revealed significant variations in mean annual rainfall and its spatial distribution across the study area. Additionally, there were notable inter-station differences in both average maximum and minimum temperatures on an annual basis.

The projected periods (2020–2051 and 2051–2081) were analyzed under the SSP2–4.5 and SSP5–8.5 climate scenarios. Compared to the baseline period, both scenarios indicate a decline in mean annual precipitation by 10% under SSP2–4.5 and 3.68% under SSP5–8.5 in the near future. The mean annual minimum temperature is projected to increase by 21.8% under SSP2–4.5 and by 22.6% under SSP5–8.5. Similarly, the mean annual maximum temperature is expected to rise by 5.55% under SSP2–4.5 and 6.33% under SSP5–8.5. These projections suggest that, in the near future, the mean annual minimum temperature in the study area will increase at a faster rate than the mean annual maximum temperature.

Moreover, the warming trend is slightly more pronounced under scenarios with higher radiative forcing. These changes in temperature and precipitation are likely to have significant implications for the region's climate, affecting agriculture, water resources, and ecosystems. Potential impacts include increased heat stress, shifts in growing seasons, and alterations in hydrological patterns.

During the mid-future period (2051–2081), under the SSP2–4.5 and SSP5–8.5 scenarios, the mean annual minimum temperature in the North Gojjam sub-basin is projected to increase by 0.31°C and 0.39°C, respectively. Similarly, the mean annual maximum temperature is expected to rise by 0.09°C under SSP2–4.5 and by 0.11°C under SSP5–8.5. Precipitation trends during the same period indicate a decrease of 5.13% under SSP2–4.5, while an increase of 2.82% is anticipated under SSP5–8.5.

The northwest and southeast regions of the sub-basin are identified as meteorological drought hotspots, making them particularly vulnerable under future climate scenarios. The study further suggests that the duration, frequency, and intensity of droughts are likely to increase in both the near and mid-future periods, especially under the high-emission SSP5–8.5 scenario. These projections imply that, due to rising temperatures and declining precipitation across much of the study area, significant drought hazards may emerge, as evidenced by declining SPEI-3 values. Moreover, the characteristics of these drought hotspots vary across scenarios, highlighting an increasing probability of prolonged dry spells and signaling considerable climate shifts in the region.

Evaluating the temporal patterns of drought occurrences, including their frequency, severity, and recurrence rates, is essential for developing effective drought risk mitigation strategies. Identifying both the frequency and magnitude of drought events is a fundamental component of drought assessment [19,93,94]. Sirdaş & Sen [95] emphasized that the overall severity of a drought is determined by cumulative water shortages, particularly the total rainfall deficit over the drought's duration.

The findings of this study indicate an increase in both the frequency and severity of drought events across the study stations when comparing the historical period to the near- and mid-future periods. Moreover, drought recurrence at the seven stations is projected to become more severe in both future periods relative to the baseline.

According to Edossa et al. [96], drought events in Ethiopia occurred once every ten to fifteen years a century ago. However, over the past three decades, this pattern has shifted, with droughts becoming more frequent, prolonged, and intense—a trend expected to continue in the near and mid-future under medium- and high-emission climate scenarios. This shift is largely attributed to declines in both the quality and intensity of rainfall.

The projected trend indicates that drought events will become increasingly severe, frequent, and difficult to manage. These escalating drought conditions are expected to have significant impacts on agriculture, which remains the primary livelihood for the majority of the population in the study area [97]. Edossa et al. [96] reported severe agricultural consequences resulting from droughts, including complete crop failures and widespread livestock mortality. Due to the recurrent nature of these droughts, their effects on agricultural productivity may persist for years.

Strengthening drought coping mechanisms and implementing effective mitigation strategies, such as improved water harvesting and sustainable water management, are critical for addressing these challenges across all agroecological zones in the region.

Mishra & Singh [98] suggest that increasing water consumption is a more significant driver of drought than population growth or the depletion of natural water sources. In the study area, drought conditions are likely exacerbated by human activities such as accelerated soil degradation and environmental deterioration, which reduce productivity. These impacts are further intensified by rapid population growth and the limited use of scientific agricultural practices. Together, these factors are expected to worsen the environmental, economic, and social consequences of drought in the region. Therefore, mitigating these impacts and improving the quality of life for local communities requires focused, coordinated efforts and targeted intervention strategies

## Implications of the study

Numerous studies have linked rising temperatures, a key aspect of global climate change, to an increase in the frequency and duration of droughts across various regions. These intensifying drought patterns could severely impact water resource availability, hinder agricultural development, and destabilize local livelihoods, ultimately presenting significant challenges to society at large.

The findings of this study are consistent with existing research and indicate a significant projected decline in precipitation within the study area. Such changes are likely to affect multiple environmental and socioeconomic sectors, including human health, water resources, energy, agriculture, and the broader economy. In Ethiopia, the evolving climate is expected to compromise both the quality and quantity of water resources, thereby impacting lakes, rivers, reservoirs, and aquifers [99,100].

Furthermore, the region is home to the Grand Ethiopian Renaissance Dam (GERD) within the Abay River Basin, a strategic infrastructure highly dependent on stable water availability. Climate variability and declining water resources are therefore expected to have significant repercussions for the energy sector, particularly hydroelectric power, which is especially vulnerable to changes in precipitation and runoff.

The agricultural sector is also projected to become increasingly vulnerable due to climate change, posing serious threats to food security. The anticipated decline in rainfall and rise in temperatures in the North Gojjam sub-basin could severely affect rainfed agriculture, leading to reductions in both crop yield and quality. Additionally, public health is expected to be negatively impacted by these climatic shifts. Rising temperatures and reduced rainfall may increase the incidence of heat-related illnesses such as dehydration and heat stress. Shifts in precipitation patterns may also contribute to the spread of vector-borne and waterborne diseases.

## Conclusions

The study offers a comprehensive evaluation of historical and projected climatic conditions in the North Gojjam sub-basin, emphasizing their implications for drought frequency, intensity, and duration. Analysis of historical data from 1985 to 2018 revealed significant regional variations in temperature and rainfall patterns. Rainfall trends varied across locations, with some areas experiencing increases and others decreases, as recorded by various weather stations. Temperature patterns showed a consistent upward trend, with minimum temperatures rising more rapidly than maximum temperatures. These changes have already led to notable disparities in drought frequency and duration, with Tis Abay recording the highest number of drought occurrences and Mekane Eyesus experiencing the longest drought durations during the baseline period.

Climate projections for the near and mid-future periods under the SSP2–4.5 and SSP5–8.5 scenarios indicate continued warming and varied precipitation trends. Minimum temperatures are projected to increase more rapidly than maximum temperatures, indicating potential shifts in local climate dynamics. While the SSP2–4.5 scenario forecasts a decline in precipitation in the near future, SSP5–8.5 anticipates increased precipitation during the mid-term period. These projections suggest significant climatic shifts that may disrupt the hydrological balance and intensify water stress in the North Gojjam sub-basin.

The analysis further indicates that both SSP2–4.5 and SSP5–8.5 scenarios are likely to increase drought frequency and severity, with more pronounced effects under SSP5–8.5. The northwestern and southeastern parts of the sub-basin are identified as drought hotspots, projected to face severe and prolonged droughts in the near and mid-term. This anticipated escalation underscores the urgent need for adaptive measures, especially in areas with fragile ecosystems and a strong dependence on rain-fed agriculture.

Climate change-induced droughts are expected to be compounded by human-driven factors such as soil erosion and unsustainable agricultural practices. In combination with rising temperatures and shifting precipitation patterns, these factors pose significant threats to agricultural productivity and water resource management in the region. The findings highlight the need for integrated strategies to mitigate the compounded effects of climate change and anthropogenic pressures.

To address these challenges, prioritizing mitigation and adaptation strategies is essential. Key measures include improving water storage systems, promoting sustainable farming practices, and implementing more effective early warning systems to enhance regional resilience. Additionally, minimizing the impact on ecosystems and

communities requires tailored approaches that reflect the unique drought characteristics of different parts of the sub-basin.

The study highlights the profound impact of climate change on agriculture, livelihoods, and water supply in the North Gojjam sub-basin. Rising temperatures, changing precipitation patterns, and increasing drought events are likely to undermine the sustainability of agricultural practices by intensifying heat stress and disrupting growing seasons. Policymakers, researchers, and stakeholders can leverage these insights to develop integrated policies aimed at mitigating climate risks and fostering sustainable development in the region.

## Supporting information

**S1 Fig.  Annual rainfall trends for seven locations between 1985 and 2018.**
(DOCX)

**S2 Fig.  Mean annual maximum Temperature trends for seven locations between 1985 and 2018.**
(DOCX)

**S3 Fig.  Mean annual minimum temperature trends for seven locations between 1985 and 2018.**
(DOCX)

**S4 Fig.  Annual rainfall trends for seven locations from 2020 to 2050 under the SSP2–4.5 climate scenario.**
(DOCX)

**S5 Fig.  Mean annual maximum temperature trends for seven locations from 2020 to 2050 under the SSP2–4.5 climate scenario.**
(DOCX)

**S6 Fig.  Mean annual minimum temperature trends for seven locations from 2020 to 2050 under the SSP2–4.5 climate scenario.**
(DOCX)

**S7 Fig.  Annual rainfall trends for seven locations from 2020 to 2050 under the SSP5–8.5 climate scenario.**
(DOCX)

**S8 Fig.  Mean annual maximum temperature trends for seven locations from 2020 to 2050 under the SSP5–8.5 climate scenario.**
(DOCX)

**S9 Fig.  Mean annual minimum temperature trends for seven locations from 2020 to 2050 under the SSP5–8.5 climate scenario.**
(DOCX)

**S10 Fig.  Annual rainfall trends for seven locations from 2051 to 2081 under the SSP2–4.5 climate scenario.**
(DOCX)

**S11 Fig.  Mean annual maximum temperature trends for seven locations from 2051 to 2081 under the SSP2–4.5 climate scenario.**
(DOCX)

**S12 Fig.  Mean annual minimum temperature trends for seven locations from 2051 to 2081 under the SSP2–4.5 climate scenario.**
(DOCX)

**S13 Fig. Annual rainfall trends for seven locations from 2051 to 2081 under the SSP5–8.5 climate scenario.**
(DOCX)

**S14 Fig. Mean annual maximum temperature trends for seven locations from 2051 to 2081 under the SSP5–8.5 climate scenario.**
(DOCX)

**S15 Fig. Mean annual minimum temperature trends for seven locations from 2051 to 2081 under the SSP5–8.5 climate scenario.**
(DOCX)

## Acknowledgments

We would like to express our gratitude to the Ethiopian National Meteorological Agency for providing the necessary climate data. Additionally, we would like to thank the ESFG website for allowing us to freely access the CMIP6 GCMs data. The authors greatly appreciate the contributions of these institutions in providing the data.

## Author contributions

**Conceptualization:** Tatek Belay.

**Data curation:** Tatek Belay.

**Formal analysis:** Tatek Belay.

**Investigation:** Tadele Melese.

**Methodology:** Tatek Belay, Tadele Melese, Baye Terefe.

**Software:** Tatek Belay, Teshager Zerihun Nigussie.

**Writing – original draft:** Tatek Belay, Tadele Melese, Baye Terefe.

**Writing – review & editing:** Tadele Melese, Baye Terefe, Simachew Bantigegn Wassie, Teshager Zerihun Nigussie.

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
