## [Decision Letter · Decision Letter 0]

PONE-D-24-44303Meteorological drought under historical and future climate scenarios in North Gojjam sub-basin, Abay River basin of EthiopiaPLOS ONE

Dear Dr. Belay,

Thank you for submitting your manuscript to PLOS ONE. After careful consideration, we feel that it has merit but does not fully meet PLOS ONE’s publication criteria as it currently stands. Therefore, we invite you to submit a revised version of the manuscript that addresses the points raised during the review process.

**ACADEMIC EDITOR: ** The manuscript needs to be improved significantly by implementing reviewers' comments. Please provide a detailed point-by-point response file in addition to the revised manuscript. Moreover, I also agree with one of the reviewer comments about using CMIP6. Indeed, adopting a recent version of climate projections can provide a better perspective. Thus, I recommend using CMIP6 instead of CMIP5 unless you have a rational reason for not using it.

We look forward to receiving your revised manuscript.

Kind regards,

Majid Niazkar, Ph.D.

Academic Editor

PLOS ONE

Reviewers' comments:

Reviewer's Responses to Questions

**Comments to the Author**

1. Is the manuscript technically sound, and do the data support the conclusions?

Reviewer #1: Yes

Reviewer #2: Yes

Reviewer #3: No

2. Has the statistical analysis been performed appropriately and rigorously? 

Reviewer #1: Yes

Reviewer #2: Yes

Reviewer #3: No

3. Have the authors made all data underlying the findings in their manuscript fully available?

Reviewer #1: Yes

Reviewer #2: No

Reviewer #3: Yes

4. Is the manuscript presented in an intelligible fashion and written in standard English?

Reviewer #1: Yes

Reviewer #2: Yes

Reviewer #3: No

5. Review Comments to the Author

Reviewer #1: I have the reviewed the manuscript entitled "Meteorological drought under historical and future climate scenarios in North Gojjam sub-basin, Abay River basin of Ethiopia". The whole paper is clear and the different sections are explained in a very detailed and clear way. The manuscript is within the scope of the journal. The methods are sound and the results are consistent with the methods. However the following needs to be addressed:

In Figure 1, the map in the top right corner is too small. It needs to be enlarged because the lettering overlaps. I would also make a clearer outline of the basin, the image below where you see the basin has the watershed cut out. It definitely needs to be redone.

Figure 2 needs to be enlarged. Also, the x-axis is truncated in almost all the subplots, and you can't read 2020 in the pictures on the right.

Overall, the manuscript is of good scientific quality.

Reviewer #2: The article "Meteorological drought under historical and future climate scenarios in North Gojjam sub-basin, Abay River basin of Ethiopia" discovers the future possibility of drought in the Gojjam sub-basin which is important for the region. The article can be improved by considering following comments.

Provide a flow chart summarising the entire methodology adopted.

Include some recent literature from different part of the world such as https://link.springer.com/article/10.1007/s11269-024-03959-w

Provide basic equations of trend test.

Please provide conclusion in point wise manner.

Reviewer #3: Introduction :

The introduction acknowledges the inherent uncertainties in climate models but does not specify how these uncertainties will be addressed or quantified. give the results of uncertainties.

While CMIP5 is a widely used dataset, it has been partially superseded by CMIP6, which includes updated scenarios and improved model simulations. Not using the latest data could limit the study's relevance to current and future research contexts.

The introduction focuses heavily on Ethiopia but lacks a broader comparison or connection to global drought trends. Adding context about how findings in North Gojjam might contribute to a global understanding of drought dynamics would strengthen the study's appeal.

While the study aims to understand meteorological droughts, it overlooks the implications for agriculture, water resources, or livelihoods. Addressing these aspects would enhance the practical relevance of the research.

Methods:

SPEI relies on the accuracy of temperature and precipitation data, which might be prone to uncertainties in data-scarce regions.

The use of a log-logistic distribution assumes that the dataset follows this distribution, which might not always hold true. Including these limitations would provide a more balanced perspective.

The equations are essential, but their presentation could be simplified. Including explanations for each variable in the text rather than relying on standalone equations would make the methodology more accessible to readers unfamiliar with the HS equation or log-logistic functions.

The explanation of how extraterrestrial radiation (Ra) is derived (e.g., through Julian days and latitude) is brief and could confuse readers. A short elaboration or reference to a supplementary figure/table detailing Ra calculation would help.

The choice of time scales for SPEI calculation (monthly, annual, multi-scale) is not explicitly provided. Why were these time scales selected, and how do they align with the study’s objectives? Addressing this would strengthen the methodological rationale.

While the methodology for calculating SPEI is not well-detailed, it is unclear how this is integrated with CMIP5 data to assess future drought conditions. A brief discussion of how SPEI calculations are adapted for future climate scenarios would bridge the gap between the methods and the study's aims.

The section uses terms like "water surplus," "water deficit," "atmospheric evaporative demand," and "water balance." While these are accurate, their relationship to each other and to SPEI calculation could be clarified for consistency.

Some sentences are dense and could benefit from rephrasing for better readability (e.g., lines 211–214).

Break down the paragraph into smaller, logically organized sections with subheadings (e.g., "SPI and Its Limitations," "SPEI Calculation," "HS Equation for PET," etc.).

6. PLOS authors have the option to publish the peer review history of their article (what does this mean? ). If published, this will include your full peer review and any attached files.

**Do you want your identity to be public for this peer review?** For information about this choice, including consent withdrawal, please see our Privacy Policy .

Reviewer #1: No

Reviewer #2: No

Reviewer #3: No

---

## [Author Response · Author response to Decision Letter 1]

28 Jan 2025

Response to editor and reviewers on “Meteorological drought under historical and future climate scenarios in North Gojjam sub-basin, Abay River basin of Ethiopia”

Manuscript Number: PONE-D-24-44303

Dear Editor

Thank you for giving me the opportunity to submit a revised draft of our manuscript titled Meteorological drought under historical and future climate scenarios in North Gojjam sub-basin, Abay River basin of Ethiopia to PLOS ONE. We appreciate the time and effort that you and the reviewers have dedicated to providing your valuable feedback on our manuscript. We are grateful to the reviewers for their insightful comments on our paper. We have been able to incorporate changes to reflect most of the suggestions provided by the reviewers. We have highlighted the changes within the manuscript.

Academic editor comments: The manuscript needs to be improved significantly by implementing reviewers' comments. Please provide a detailed point-by-point response file in addition to the revised manuscript. Moreover, I also agree with one of the reviewer comments about using CMIP6. Indeed, adopting a recent version of climate projections can provide a better perspective. Thus, I recommend using CMIP6 instead of CMIP5 unless you have a rational reason for not using it.

Response: Dear Academic Editor,

Thank you for your constructive comments and suggestions, as well as for considering this manuscript for publication in the high-impact journal PLOS ONE. Based on your and the reviewers' feedback, we have made the following revisions:

• We have replaced CMIP5 with CMIP6.

• The number of stations used in the analysis has been updated to seven due to similarities in the minimum and maximum temperature and precipitation data extracted from the stations using GCMs. As a result of these changes, the analysis section has also been revised to reflect the use of CMIP6.

Comment 2: Please include the following items when submitting your revised manuscript:

Please reference all numbered equations in text. Currently, numbered equations [1] in the manuscript have not been cited in text.

Response: Dear editor, Thank you for your comments. We have addressed the points raised by the academic editor and reviewers, and our detailed responses are attached as a separate file for your reference. Additionally, we have submitted both a clean version of the manuscript (without tracked changes) and a revised version with tracked changes, as per your request. All equations have now been properly cited within the text.

Editor and Reviewer comments:

Comment 1: Journal requirements: When submitting your revision, we need you to address these additional requirements. Please ensure that your manuscript meets PLOS ONE's style requirements, including those for file naming.

Response 1: Thank you for your comments. We have reviewed the requirements of PLOS ONE, including those for file names, and have made corrections accordingly to ensure compliance with the journal's guidelines.

Comment 2: When completing the data availability statement of the submission form, you indicated that you will make your data available on acceptance. We strongly recommend all authors decide on a data sharing plan before acceptance, as the process can be lengthy and hold up publication timelines. Please note that, though access restrictions are acceptable now, your entire data will need to be made freely accessible if your manuscript is accepted for publication. This policy applies to all data except where public deposition would breach compliance with the protocol approved by your research ethics board. If you are unable to adhere to our open data policy, please kindly revise your statement to explain your reasoning and we will seek the editor's input on an exemption. Please be assured that, once you have provided your new statement, the assessment of your exemption will not hold up the peer review process.

Response 2: Thank you for your comments. I acknowledge and accept the PLOS ONE data policy and agree to share the data used in this study upon acceptance of the manuscript.

Comment 3. We note that Figure 1 in your submission contain [map/satellite] images which may be copyrighted. All PLOS content is published under the Creative Commons Attribution License (CC BY 4.0), which means that the manuscript, images, and Supporting Information files will be freely available online, and any third party is permitted to access, download, copy, distribute, and use these materials in any way, even commercially, with proper attribution. For these reasons, we cannot publish previously copyrighted maps or satellite images created using proprietary data, such as Google software (Google Maps, Street View, and Earth). For more information, see our copyright guidelines: http://journals.plos.org/plosone/s/licenses-and-copyright. We require you to either (1) present written permission from the copyright holder to publish these figures specifically under the CC BY 4.0 license, or (2) remove the figures from your submission:

Response 3: Thank you for your comments. The map presented in Figure 1 is original and has not been published elsewhere. It was created specifically for this study using publicly available and open-source data. I confirm that it fully complies with the requirements of the Creative Commons Attribution License (CC BY 4.0).

Reviewers' comments:

Reviewer's Responses to Questions

Comments to the Author

1. Is the manuscript technically sound, and do the data support the conclusions?

Reviewer #1: Yes

Reviewer #2: Yes

Reviewer #3: No

Response 1: Dear Reviewers, we sincerely thank you for your time, appreciation, and constructive comments, which have been invaluable for improving the manuscript.

2. Has the statistical analysis been performed appropriately and rigorously?

Reviewer #1: Yes

Reviewer #2: Yes

Reviewer #3: No

Response 2: Dear reviewers, we greatly appreciate your time, thoughtful feedback, and constructive comments, which have significantly contributed to enhancing the manuscript.

3. Have the authors made all data underlying the findings in their manuscript fully available?

Reviewer #1: Yes

Reviewer #2: No

Reviewer #3: Yes

Response 3: Dear Reviewers, thank you for your valuable time and constructive feedback to improve the manuscript.

4. Is the manuscript presented in an intelligible fashion and written in standard English?

Reviewer #1: Yes

Reviewer #2: Yes

Reviewer #3: No

Response 4: Dear Reviewers, thank you for your valuable time and constructive feedback to improve the manuscript.

5. Review Comments to the Author

Please use the space provided to explain your answers to the questions above. You may also include additional comments for the author, including concerns about dual publication, research ethics, or publication ethics. (Please upload your review as an attachment if it exceeds 20,000 characters).

Comments from Reviewer 1#

Comment 1: I have the reviewed the manuscript entitled "Meteorological drought under historical and future climate scenarios in North Gojjam sub-basin, Abay River basin of Ethiopia". The whole paper is clear and the different sections are explained in a very detailed and clear way. The manuscript is within the scope of the journal. The methods are sound and the results are consistent with the methods. However the following needs to be addressed: In Figure 1, the map in the top right corner is too small. It needs to be enlarged because the lettering overlaps. I would also make a clearer outline of the basin, the image below where you see the basin has the watershed cut out. It definitely needs to be redone.

Figure 2 needs to be enlarged. Also, the x-axis is truncated in almost all the subplots, and you can't read 2020 in the pictures on the right. Overall, the manuscript is of good scientific quality.

Response 1: Thank you for taking the time to review our manuscript and for your positive feedback. We have revised Figures 1 and 2 in the updated manuscript to enhance their clarity and visibility (see the maps).

Reviewer #2: The article "Meteorological drought under historical and future climate scenarios in North Gojjam sub-basin, Abay River basin of Ethiopia" discovers the future possibility of drought in the Gojjam sub-basin which is important for the region. The article can be improved by considering following comments.

Provide a flow chart summarising the entire methodology adopted.

Include some recent literature from different part of the world such as https://link.springer.com/article/10.1007/s11269-024-03959-w

Provide basic equations of trend test. Please provide conclusion in point wise manner.

Response 2: Dear reviewer, Thank you for your valuable comment and suggestion. We have now included a summary of the entire methodology in the revised manuscript based on your suggestion. Please refer to the updated version for details.

Reviewer #3: Introduction:

Comment #1: The introduction acknowledges the inherent uncertainties in climate models but does not specify how these uncertainties will be addressed or quantified. give the results of uncertainties.

Response 1: Thank you for your comment. We have now revised the introduction section in the updated manuscript.

Comment #2: While CMIP5 is a widely used dataset, it has been partially superseded by CMIP6, which includes updated scenarios and improved model simulations. Not using the latest data could limit the study's relevance to current and future research contexts.

Response 2: Dear reviewer, Thank you for your valuable comments. We have now used CMIP6 instead of CMIP5, which has resulted in modifications to the analysis section of the manuscript based on the updated data. The comments have been addressed and incorporated into the revised manuscript. Please find the updated version for your review.

Comment #3: The introduction focuses heavily on Ethiopia but lacks a broader comparison or connection to global drought trends. Adding context about how findings in North Gojjam might contribute to a global understanding of drought dynamics would strengthen the study's appeal.

While the study aims to understand meteorological droughts, it overlooks the implications for agriculture, water resources, or livelihoods. Addressing these aspects would enhance the practical relevance of the research.

Response 3: Dear reviewer, Thank you for your comment. The implications of the study for agricultural water resource management and livelihoods have been incorporated into the updated manuscript.

Methods:

SPEI relies on the accuracy of temperature and precipitation data, which might be prone to uncertainties in data-scarce regions. The use of a log-logistic distribution assumes that the dataset follows this distribution, which might not always hold true. Including these limitations would provide a more balanced perspective. The equations are essential, but their presentation could be simplified. Including explanations for each variable in the text rather than relying on standalone equations would make the methodology more accessible to readers unfamiliar with the HS equation or log-logistic functions. The explanation of how extraterrestrial radiation (Ra) is derived (e.g., through Julian days and latitude) is brief and could confuse readers. A short elaboration or reference to a supplementary figure/table detailing Ra calculation would help. The choice of time scales for SPEI calculation (monthly, annual, multi-scale) is not explicitly provided. Why were these time scales selected, and how do they align with the study’s objectives? Addressing this would strengthen the methodological rationale. While the methodology for calculating SPEI is not well-detailed, it is unclear how this is integrated with CMIP5 data to assess future drought conditions. A brief discussion of how SPEI calculations are adapted for future climate scenarios would bridge the gap between the methods and the study's aims.

Response: Thank yo

---

## [Decision Letter · Decision Letter 1]

PONE-D-24-44303R1Meteorological drought under historical and future climate scenarios in North Gojjam sub-basin, Abay River basin of EthiopiaPLOS ONE

Dear Dr. Belay,

Thank you for submitting your manuscript to PLOS ONE. After careful consideration, we feel that it has merit but does not fully meet PLOS ONE’s publication criteria as it currently stands. Therefore, we invite you to submit a revised version of the manuscript that addresses the points raised during the review process.

**ACADEMIC EDITOR: **The manuscript still needs to be signifincally improved as one of the reviewers still suggested rejection. When submitting your revised manuscript, provide a point-by-point response to all review comments so that a suitable decision is made for your artcile. 

We look forward to receiving your revised manuscript.

Kind regards,

Majid Niazkar, Ph.D.

Academic Editor

PLOS ONE

**Additional Editor Comments:**

Since their response did not convience one of the reviewers, we invited another expert in the field to evalute the work. The auhtors need to provide point-by-point reponse to the review comments. Morevoer, if any reviewer suggested citing a paper, you need to check if citing the suggested reference is improving your work or not. You can add the reference only if it improves your work. Otherwise, please provide a response to the raised comment.

Reviewers' comments:

Reviewer's Responses to Questions

**Comments to the Author**

1. If the authors have adequately addressed your comments raised in a previous round of review and you feel that this manuscript is now acceptable for publication, you may indicate that here to bypass the “Comments to the Author” section, enter your conflict of interest statement in the “Confidential to Editor” section, and submit your "Accept" recommendation.

Reviewer #3: All comments have been addressed

Reviewer #4: All comments have been addressed

2. Is the manuscript technically sound, and do the data support the conclusions?

Reviewer #3: Partly

Reviewer #4: Yes

3. Has the statistical analysis been performed appropriately and rigorously? 

Reviewer #3: No

Reviewer #4: Yes

4. Have the authors made all data underlying the findings in their manuscript fully available?

Reviewer #3: No

Reviewer #4: Yes

5. Is the manuscript presented in an intelligible fashion and written in standard English?

Reviewer #3: Yes

Reviewer #4: Yes

6. Review Comments to the Author

Reviewer #3: Despite the modifications made by the researchers, I believe the study requires greater depth to justify its publication, particularly by examining the resulting effects. The analysis should go beyond assessing atmospheric drought over a period of more than thirty years, as trends have been shown to vary with the length of the time series, making generalization insufficient

Reviewer #4: PLOS ONE

Meteorological drought under historical and future climate scenarios in North Gojjam

sub-basin, Abay River basin of Ethiopia

Dear Editor

This manuscript assessed meteorological drought in the North Gojjam sub-basin of Ethiopia using the SPEI index and run theory, analyzing past and future trends in precipitation and temperature. Results show a sharper rise in minimum temperature and an increase in drought frequency and severity. The northeastern and southeastern areas are most vulnerable, highlighting the importance of temporal drought forecasting for effective water resource management.

I recommend that this manuscript can be accepted.

Sincerely,

I here summarize these comments:

Comments:

1. In the Materials and Methods section, explain more about the Mann-Kendall method of routing and provide the relevant relationships. You can use the following resources for this.

https://doi.org/10.1016/B978-0-323-91910-4.00032-7

https://doi.org/10.3390/w16213077

2. There are two tables with the number 3!

3. The format of Table 1 and 3 is not consistent with other tables.

4. To make things better, you can also calculate the uncertainty of temperature and precipitation data obtained from climate change scenarios, as in the reference below.

https://doi.org/10.3390/hydrology11010002

5. The abstract is too long, please summarize it.

6. In the section on calculating the SPEI index, show how to obtain this index in the form of relationships so that the reader can understand this topic at a glance

https://dor.isc.ac/dor/20.1001.1.17352347.1398.15.4.24.7

7. It would be better to also show past temperature and precipitation data and drought index in graphical form so that the reader can get a better understanding of the data trends.

8. Lines 163 to 169 require references. You can use the following references.

https://doi.org/10.1002/clen.201100652

https://doi.org/10.3390/hydrology9100176

7. PLOS authors have the option to publish the peer review history of their article (what does this mean? ). If published, this will include your full peer review and any attached files.

**Do you want your identity to be public for this peer review?** For information about this choice, including consent withdrawal, please see our Privacy Policy .

Reviewer #3: No

Reviewer #4: No

---

## [Author Response · Author response to Decision Letter 2]

22 Apr 2025

Response to editor and reviewers on “Meteorological drought under historical and future climate scenarios in North Gojjam sub-basin, Abay River basin of Ethiopia”

Manuscript Number: PONE-D-24-44303R1

Dear Editor: PLOS ONE

Thank you for giving us the opportunity to submit a revised draft of our manuscript titled Meteorological drought under historical and future climate scenarios in North Gojjam sub-basin, Abay River basin of Ethiopia to PLOS ONE.

We appreciate the efforts that you and the reviewers have made to provide valuable feedback and suggestions to improve our manuscript. We are grateful to the reviewers for their insightful comments on the paper. We are delighted to have been able to respond to the suggestions provided by the reviewers. Below are point-by-point responses to the reviewers' comments. The reviewers' comments are in Italic font, and our responses are in turquoise.

N.B: In the revised manuscript, only the major corrections are highlighted in turquoise.

The text highlighted in turquoise reflects changes made in response to the comments and suggestions of the reviewer

Editor comments:

Since their response did not convience one of the reviewers, we invited another expert in the field to evalute the work. The authors need to provide point-by-point reponse to the review comments. Moreover, if any reviewer suggested citing a paper, you need to check if citing the suggested reference is improving your work or not. You can add the reference only if it improves your work. Otherwise, please provide a response to the raised comment.

Response:

Dear editor,

Thank you for your comments. We have addressed all points raised by the academic editor and reviewers. A detailed point-by-point response is provided in the attached file. As requested, we have submitted both a clean and a tracked-changes version of the manuscript. Suggested references have been incorporated where relevant to the manuscript.

Academic editor comments: The manuscript still needs to be signifincally improved as one of the reviewers still suggested rejection. When submitting your revised manuscript, provide a point-by-point response to all review comments so that a suitable decision is made for your article.

Response: Dear Academic Editor, Thank you for the feedback. We have substantially revised the manuscript and addressed all reviewer comments in detail. A point-by-point response is provided to facilitate a thorough re-evaluation.

Reviewer comments:

# 1. In the Materials and Methods section, explain more about the Mann-Kendall method of routing and provide the relevant relationships. You can use the following resources for this.

https://doi.org/10.1016/B978-0-323-91910-4.00032-7

https://doi.org/10.3390/w16213077

Response 1: Thank you for your valuable comments. We have now expanded the Materials and Methods section to provide a more detailed explanation of the Mann-Kendall method. In line with your suggestion, we have also included and cited the recommended source. Please refer to the revised section in the updated manuscript for the incorporation of your comments.

#2. There are two tables with the number 3

Response 2: Thank you for your constructive comment. We have corrected the issue of duplicate table numbering, and the revised manuscript now reflects the appropriate table numbers.

#3. The format of Table 1 and 3 is not consistent with other tables.

Response 3: Thank you for your valuable comment. We have carefully reviewed and revised the formatting of Tables 1 and 3 to ensure consistency with the other tables in the revised manuscript

# 4. To make things better, you can also calculate the uncertainty of temperature and precipitation data obtained from climate change scenarios, as in the reference below.

https://doi.org/10.3390/hydrology11010002

Response: Thank you very much for your valuable suggestion regarding the calculation of uncertainty in temperature and precipitation data derived from climate change scenarios. We fully agree that uncertainty analysis is a crucial aspect of climate modeling. However, in the present study, we addressed model reliability and robustness using a comprehensive set of statistical validation metrics, including the correlation coefficient (R), Percent Bias (PBIAS), and Root Mean Square Error (RMSE). These methods were carefully selected to evaluate the model's performance across different scenarios.

#5. The abstract is too long, please summarize it.

Response: Thank you for your comment. We have revised and shortened the abstract to provide a more concise summary of the manuscript in the revised version.

#6. In the section on calculating the SPEI index, show how to obtain this index in the form of relationships so that the reader can understand this topic at a glance

https://dor.isc.ac/dor/20.1001.1.17352347.1398.15.4.24.7

Response: Thank you for your constructive comment and valuable suggestion. In response, we have revised the manuscript to include a clear and concise explanation of the SPEI calculation process. The step-by-step formulation is now presented using mathematical expressions to help readers easily understand how the index is derived. Please refer to the updated manuscript for these revisions.

7. It would be better to also show past temperature and precipitation data and drought index in graphical form so that the reader can get a better understanding of the data trends.

Response: Thank you for your valuable suggestion. We have included temperature and precipitation trends in graphical form as supplementary figures. Due to the large number of figures already present in the manuscript, we have placed these additional graphs in the supplementary materials (S1 Fig. 1–S15 Fig.), which are included in the revised manuscript.

8. Lines 163 to 169 require references. You can use the following references.

https://doi.org/10.1002/clen.201100652

https://doi.org/10.3390/hydrology9100176

Response: Thank you for your valuable suggestion. We have revised the manuscript and incorporated the recommended reference to support Lines 120–126. The updated citations are now included on page 5, starting at line 126.

Thank you once again for your valuable feedback and for helping to improve our manuscript.

In addition to addressing the comments above, we have corrected all spelling and grammatical errors. Furthermore, as suggested by the editor, we have included the manuscript highlights.

We look forward to hearing from you in due course regarding our submission and are happy to respond to any further questions or comments you may have.

Thank you for your valuable and constructive comments and suggestions!

---

## [Decision Letter · Decision Letter 2]

PONE-D-24-44303R2Meteorological drought under historical and future climate scenarios in North Gojjam sub-basin, Abbay River basin of EthiopiaPLOS ONE

Dear Dr. Belay,

Thank you for submitting your manuscript to PLOS ONE. After careful consideration, we feel that it has merit but does not fully meet PLOS ONE’s publication criteria as it currently stands. Therefore, we invite you to submit a revised version of the manuscript that addresses the points raised during the review process.

**ACADEMIC EDITOR: The authors need to provide a point-by-point response to the minor comments raised by the reviewers.**==============================

We look forward to receiving your revised manuscript.

Kind regards,

Majid Niazkar, Ph.D.

Academic Editor

PLOS ONE

Journal Requirements:

Reviewers' comments:

Reviewer's Responses to Questions

**Comments to the Author**

1. If the authors have adequately addressed your comments raised in a previous round of review and you feel that this manuscript is now acceptable for publication, you may indicate that here to bypass the “Comments to the Author” section, enter your conflict of interest statement in the “Confidential to Editor” section, and submit your "Accept" recommendation.

Reviewer #4: All comments have been addressed

Reviewer #5: (No Response)

2. Is the manuscript technically sound, and do the data support the conclusions?

Reviewer #4: Yes

Reviewer #5: Yes

3. Has the statistical analysis been performed appropriately and rigorously? 

Reviewer #4: Yes

Reviewer #5: Yes

4. Have the authors made all data underlying the findings in their manuscript fully available?

Reviewer #4: Yes

Reviewer #5: Yes

5. Is the manuscript presented in an intelligible fashion and written in standard English?

Reviewer #4: Yes

Reviewer #5: Yes

6. Review Comments to the Author

Reviewer #4: Dear Editor

This version of the article has been revised.

All corrections have been made. Therefore, the article is accepted.

Reviewer #5: I recommend that the authors carefully address the following minor technical and editorial points:

Minor Revision:

1- Line 2: “...climate scenarios in North Gojjam sub-basin, Abay River basin of Ethiopia”

Use consistent spelling “Abay River Basin” (as used in the title). Standardize throughout the text.

2- Add a space between the number and unit: 24.7°C → 24.7 °C.

3- Line 218: Mention the specific packages used.

4- Line 355: “...where nnn covers the entire period.” Likely typo error.

5- Tables are not formatting well such as Table 8, 9 and 10, etc.

6- Line 566: “Run theory” is mentioned briefly. Consider a 1–2 sentence clarification or citation since not all readers may be familiar with it.

7- Lines 358–365: You state autocorrelation correction is applied but don't mention the lag selection or whether pre-whitening was used.

7. PLOS authors have the option to publish the peer review history of their article (what does this mean? ). If published, this will include your full peer review and any attached files.

**Do you want your identity to be public for this peer review?** For information about this choice, including consent withdrawal, please see our Privacy Policy .

Reviewer #4: **Yes: ** Goodarzi.M.R

Reviewer #5: No

---

## [Author Response · Author response to Decision Letter 3]

27 May 2025

Response to Editor and Reviewers

Manuscript ID: PONE-D-24-44303R3

Title: Meteorological drought under historical and future climate scenarios in North Gojjam sub-basin, Abay River basin of Ethiopia

Dear Editor and Reviewers

Thank you for the opportunity to revise and resubmit our manuscript to PLOS ONE. We would like to thank for the constructive feedback provided by the reviewers and the editorial team, which has helped us improve the clarity, rigor, and impact of our study. In this revised submission, we have addressed all editorial requirements and responded to each reviewer’s comments in detail. We appreciate the efforts that you and the reviewers have made to provide valuable feedback and suggestions to improve our manuscript. We are grateful to the reviewers for their insightful comments on the paper. We are delighted to have been able to respond to the suggestions provided by the reviewers. Below are point-by-point responses to the reviewers' comments. The reviewers' comments are in Italic font. The sections highlighted in turquoise represent our responses to both the editor's and reviewers' feedback and indicate the corresponding revisions made in the main manuscript. All suggested changes have been incorporated into the revised manuscript, and we have uploaded all additional files as requested.

Response to Editorial Requirements:

We've checked your submission and before we can proceed, we need you to address the following issues:

We note that Figure 1 in your submission contain [map/satellite] images which may be copyrighted. All PLOS content is published under the Creative Commons Attribution License (CC BY 4.0), which means that the manuscript, images, and Supporting Information files will be freely available online, and any third party is permitted to access, download, copy, distribute, and use these materials in any way, even commercially, with proper attribution. For these reasons, we cannot publish previously copyrighted maps or satellite images created using proprietary data, such as Google software (Google Maps, Street View, and Earth). For more information, see our copyright guidelines: http://journals.plos.org/plosone/s/licenses-and-copyright.

Please upload the completed Content Permission Form or other proof of granted permissions as an ""Other"" file with your submission. In the figure caption of the copyrighted figure, please include the following text: “Reprinted from [ref] under a CC BY license, with permission from [name of publisher], original copyright [original copyright year].”

USGS National Map Viewer (public domain): http://viewer.nationalmap.gov/viewer/.

Response: Thank you for your detailed and thoughtful feedback on our manuscript. We sincerely appreciate your careful review and your dedication to upholding high standards and open access principles.

In response to your concern regarding Figure 1, we would like to clarify that this figure is an original map created by the authors. The underlying shapefile data used to develop the map were freely downloaded from OpenDataSoft, a public open data platform that provides spatial datasets for unrestricted use. As GIS and remote sensing experts, we routinely generate our own maps and figures using openly licensed, non-proprietary datasets, and we adhere to accepted cartographic and licensing standards.

To ensure full compliance with the Creative Commons Attribution (CC BY 4.0) license, we have updated the figure caption to clearly indicate the data source and confirm that the map does not incorporate any proprietary or copyrighted content (e.g., Google Maps or Google Earth). The revised caption now reads:

Fig 1. Map of the study area developed by the authors. Source: Original shapefile data freely downloaded from https://public.opendatasoft.com.

# Comment: Journal Requirements:

Response: Thank you for your suggestions. As per the journal’s requirements, we have thoroughly reviewed all the references. The reference list is complete and accurate, and we confirm that no retracted papers have been cited.

Reviewer comments:

Reviewer 5:

# 1. Line 2: “...climate scenarios in North Gojjam sub-basin, Abay River basin of Ethiopia. Use consistent spelling “Abay River Basin” (as used in the title). Standardize throughout the text.

Response 1: Thank you for your valuable comments. We have made the necessary corrections and ensured consistent spelling throughout the text (please refer the updated manuscript).

#2. Add a space between the number and unit: 24.7°C → 24.7 °C.

Response 2: Thank you for your constructive comment. We have addressed the issue and made the necessary corrections in the revised manuscript.

#3. Line 218: Mention the specific packages used.

Response 3: We appreciate your insightful comment and have now included the specific packages used in the revised manuscript.

# 4. Line 355: “...where nnn covers the entire period.” Likely typo error.

Response: Thank you for your valuable comment. We have corrected the typographic error.

#5. Tables are not formatting well such as Table 8, 9 and 10, etc.

Response: Thank you for your comment. We have reviewed and corrected the formatting issues in the tables, including Tables 8, 9, and 10, to ensure consistency and clarity.

#6. Line 566: “Run theory” is mentioned briefly. Consider a 1–2 sentence clarification or citation since not all readers may be familiar with it.

Response: Thank you for your constructive comment and valuable suggestion. We have now incorporated a clearer explanation of run theory to enhance understanding for all readers. Please refer to the updated manuscript.

7. Lines 358–365: You state autocorrelation correction is applied but don't mention the lag selection or whether pre-whitening was used.

Response: Thank you for pointing this out. In the revised manuscript, we have clarified that we applied a lag-1 autocorrelation correction, which is commonly used in drought and climate trend analyses.

We believe these revisions address all concerns and enhance the manuscript’s quality. Thank you again for your guidance. We look forward to your feedback!

Sincerely,

Dr. Tatek Belay

Corresponding Author

Debre Tabor University

---

## [Decision Letter · Decision Letter 3]

Meteorological drought under historical and future climate scenarios in North Gojjam sub-basin, Abay River basin of Ethiopia

PONE-D-24-44303R3

Dear Dr. Belay,

We’re pleased to inform you that your manuscript has been judged scientifically suitable for publication and will be formally accepted for publication once it meets all outstanding technical requirements.

Kind regards,

Majid Niazkar, Ph.D.

Academic Editor

PLOS ONE

Additional Editor Comments (optional):

Based on the review assessment, the manuscript can be accepted for publication.

Reviewers' comments:

Reviewer's Responses to Questions

**Comments to the Author**

1. If the authors have adequately addressed your comments raised in a previous round of review and you feel that this manuscript is now acceptable for publication, you may indicate that here to bypass the “Comments to the Author” section, enter your conflict of interest statement in the “Confidential to Editor” section, and submit your "Accept" recommendation.

Reviewer #5: All comments have been addressed

Reviewer #6: All comments have been addressed

2. Is the manuscript technically sound, and do the data support the conclusions?

Reviewer #5: Yes

Reviewer #6: Yes

3. Has the statistical analysis been performed appropriately and rigorously? 

Reviewer #5: Yes

Reviewer #6: Yes

4. Have the authors made all data underlying the findings in their manuscript fully available?

Reviewer #5: Yes

Reviewer #6: Yes

5. Is the manuscript presented in an intelligible fashion and written in standard English?

Reviewer #5: Yes

Reviewer #6: Yes

6. Review Comments to the Author

Reviewer #5: Minor Revision:

1- Add a space between the number and unit such as lines 363, 364, 373 and 374 and throughout the manuscript: 8.2°C to 12.6°C → 8.2 °C to 12.6 °C.

Reviewer #6: I have read the paper along with all the revisions, and I believe the current version is sound good and can be accepted as it is.

7. PLOS authors have the option to publish the peer review history of their article (what does this mean? ). If published, this will include your full peer review and any attached files.

**Do you want your identity to be public for this peer review?** For information about this choice, including consent withdrawal, please see our Privacy Policy .

Reviewer #5: No

Reviewer #6: No

---

## [Editor Report · Acceptance letter]

PONE-D-24-44303R3

PLOS ONE

Dear Dr. Belay,

I'm pleased to inform you that your manuscript has been deemed suitable for publication in PLOS ONE. Congratulations! Your manuscript is now being handed over to our production team.

Kind regards,

on behalf of

Dr. Majid Niazkar

Academic Editor

PLOS ONE